# *Streptomyces* alleviate abiotic stress in plant by producing pteridic acids

Zhijie Yang [1], Yijun Qiao [1], Naga Charan Konakalla [2], Emil Strøbech[1], Pernille Harris [3], Gundela Peschel[4], Miriam Agler-Rosenbaum[4], Tilmann Weber [5], Erik Andreasson [2] & Ling Ding [1] ✉

Soil microbiota can confer fitness advantages to plants and increase crop resilience to drought and other abiotic stressors. However, there is little evidence on the mechanisms correlating a microbial trait with plant abiotic stress tolerance. Here, we report that *Streptomyces* effectively alleviate drought and salinity stress by producing spiroketal polyketide pteridic acid H (**1**) and its isomer F (**2**), both of which promote root growth in *Arabidopsis* at a concentration of 1.3 nM under abiotic stress. Transcriptomics profiles show increased expression of multiple stress responsive genes in *Arabidopsis* seedlings after pteridic acids treatment. We confirm in vivo a bifunctional biosynthetic gene cluster for pteridic acids and antimicrobial elaiophylin production. We propose it is mainly disseminated by vertical transmission and is geographically distributed in various environments. This discovery reveals a perspective for understanding plant-*Streptomyces* interactions and provides a promising approach for utilising beneficial *Streptomyces* and their secondary metabolites in agriculture to mitigate the detrimental effects of climate change.

According to the Food and Agriculture Organization of the United Nations, climate change generates considerable uncertainty about future water availability in many regions. Increased water scarcity under climate change will present a major challenge for climate adaptation, while sea-level rise will affect the salinity of surface and groundwater in coastal areas. Stress caused by climate change has led to increased agricultural losses and threatened global food security[1]. Drought is considered the most damaging environmental stress, which directly affects the entire growth period of plant seeds from germination to final fruiting[2]. Drought stress can lead to increased plant osmotic regulators, the inhibition of photosynthesis, and the change of plant endogenous hormone content[3–5]. Drought stress also induces reactive oxygen species, such as superoxide radicals, hydrogen peroxide, and hydroxyl radicals, leading to oxidative stress[6]. Crop loss due to soil salinisation is another increasing threat to agriculture worldwide, which is more severe in agricultural land in coastal and arid regions[7]. Irrigation with saline water, low precipitation, and high evapotranspiration are key factors of the rapid salinisation of agricultural land[8]. These abiotic stresses of drought and salinity have brought unprecedented challenges to the development of crop farming. Compared to the heavy use of chemical fertilisers, the use of plant growth-promoting bacteria to improve plant growth under drought and salinity environments is more sustainable and gaining more attention[9,10].

Soil microbial communities are critical to plant health and their resistance to both biotic and abiotic stressors, such as pathogens, drought, salinity, and heavy metal pollution[11]. A few studies have demonstrated that many beneficial soil bacteria harbour plant growth-

[1]Department of Biotechnology and Biomedicine, Technical University of Denmark, Søltofts Plads, Building 221, 2800 Kgs Lyngby, Denmark. [2]Department of Plant Protection Biology, Swedish University of Agricultural Sciences, Sundsvägen 14, SE-230 53 Alnarp, Sweden. [3]Department of Chemistry, Technical University of Denmark, Søltofts Plads, Building 206, 2800 Kgs Lyngby, Denmark. [4]Leibniz Institute for Natural Product Research and Infection Biology—Hans Knöll Institute (HKI), Beutenbergstr. 11a, 07745 Jena, Germany. [5]The Novo Nordisk Foundation Center for Biosustainability, Technical University of Denmark, Kemitorvet, Building 220, 2800 Kgs Lyngby, Denmark. ✉e-mail: lidi@dtu.dk

promoting activities, *e.g.* by helping plants with disease suppression[12], nutrient acquisition[13], phosphorus uptake[14] and nitrogen fixations[15]. Beneficial root microbiota also regulate biosynthetic pathways in the plant itself, leading to differential alterations in the plant metabolome in response to stresses[16]. *Streptomyces* are Gram-positive filamentous bacteria, widely distributed in soil and marine environments. While they have long been considered the richest source of bioactive secondary metabolites[17], *Streptomyces* have recently drawn attention as a class of plant growth-promoting bacteria that help plants respond to adversity stress[18]. The growing evidence showed that *Streptomyces* can promote plant growth or tolerance to stressors in direct or indirect ways, by secreting plant growth regulator auxin (indole-3-acetic acid, IAA) and siderophores, inducing systemic resistance in plants, and regulating the rhizosphere microbiome via producing antibacterial compounds or signalling molecules[19–21]. Notably, the commercial product Actinovate® and Mycostop® are two *Streptomyces*-based formulations that have been widely used to suppress a wide range of diseases in a variety of crop groups as a biological fungicide/bactericide for the long term. Recently, the enrichment of *Streptomyces* has also been shown to play a subsequent role in the drought/salt tolerance of plants[22]. Despite the widespread claims of efficacy of inoculation of plant growth-promoting *Streptomyces*, the molecular basis of the growth-promoting effects and the key role of secondary/specialised metabolites in this process are largely unknown.

Here, we report that *Streptomyces iranensis* HM 35 has profound beneficial effects on helping barley alleviate osmotic, drought and salinity stress. The active components were identified as bioactive spiroketal polyketides pteridic acids H (**1**) and its isomer F (**2**) through large-scale fermentation and bioactivity-directed purification followed by NMR, MS, and X-ray crystallography. The abiotic stress mitigating effects of pteridic acids H and F have been confirmed on the model plant *Arabidopsis thaliana*, where it effectively reversed both drought and salinity stress as phytohormone-like small biomolecules at concentrations as low as 0.5 ng mL$^{-1}$ (1.3 nM). RNA sequencing results suggested that pteridic acids may assist plants in stress resistance via activating photosynthesis and regulating multiple stress response genes. Moreover, the Biosynthetic Gene Cluster (BGC) of pteridic acids (*pta*) was identified and analysed in silico, and functionally confirmed by in vivo CRISPR-based genome editing. We have furthermore conducted a survey of 81 potential producers of pteridic acids, which are widely distributed around the world. Phylogenetic and comparative genomic analysis of *pta*-containing streptomycetes suggested that these strains have evolutionary convergence in disseminating *pta* BGC through main vertical transmission and occasional horizontal gene transfer. In summary, we reveal a strategy of *Streptomyces* to secret plant growth regulators that help plants cope with abiotic stresses, which is a promising alternative solution for plant development and crop yields under the current climate change-induced environmental stresses.

## Results

### Abiotic stress-mitigating activities exhibited by *S. iranensis*
The promotion of barley growth induced by *S. iranensis* was tested under multiple abiotic stresses including osmotic, salinity and drought. The osmotic stress experiment was simulated using soils supplemented with 20% (w/v) PEG-6000 by transiently reducing the water potential of the plant. We found that *S. iranensis* played a significant role in alleviating osmotic stress in barley seedlings. The treated seedlings showed a significant increase in height, fresh weight, and dry weight compared to the control group without any extra treatment (Fig. 1a). The culture broth of *S. iranensis* also showed considerable activity for the growth of barley in alleviating salinity stress mediated by 100 mM NaCl (Fig. 1b), while *S. iranensis* was not significantly enriched in the soil around the roots of barley seedlings (Supplementary Fig. 1). Additionally, based on the analysis of barley

seedling phenotypes, the treatment with *S. iranensis* resulted in a significant improvement in plant growth recovery from drought stress (Fig. 1c). Surprisingly, *S. iranensis* promoted the growth of barley seedlings even in non-stress growth condition and thus indicate that *S. iranensis* may have potential for use as biostimulant (Supplementary Fig. 2).

### Genomic and metabolomic profiles of *S. iranensis*
To reveal the potential bioactive components, we first annotated BGCs responsible for the biosynthesis of secondary metabolites in *S. iranensis* using antiSMASH 6.0[23]. Genome sequence analysis of *S. iranensis* revealed the presence of 47 putative secondary metabolites BGCs with a variety of biosynthetic categories (Supplementary Table 1). The clusters 3, 6, 7, 8, 23, 31, 35 and 40 were annotated to have greater than 80% similarity with BGCs responsible for the biosynthesis of coelichelin, azalomycin, nigericin, elaiophylin, desferrioxamin B, ectoine, rapamycin and hygrocin, respectively. Their corresponding products were also detected and identified through High-resolution Liquid Chromatography-tandem Mass Spectrometry (HR-LC-MS/MS) as well as Global Natural Products Social (GNPS) molecular networking (Fig. 1d, Supplementary Fig. 3)[24]. However, a large number of metabolites from *S. iranensis* are still unknown. Since none of the previously identified compounds have been associated with mitigating abiotic stress in plants, we were prompted to expand the fermentation process and identify the potential bioactive compounds.

### Characterisation of the bioactive compound pteridic acid
To uncover the bioactive components, fermentation of *S. iranensis* was scaled up to 175 litres and the culture broth was subjected to separation through open-column chromatography on Amberchrom CG161Me resin, silica gel, and Sephadex LH-20. Bioactivity-guided fractionation led to the isolation of bioactive compound **1** (15.0 mg) together with its isomer compound **2** (4.0 mg) (Fig. 1e).

The bioactive component compound **1** was isolated as a white solid. Its formula of $C_{21}H_{34}O_6$ was deduced by *m/z* 383.2439 [M + H]$^+$ (calculated for 383.2428, $\Delta$ 2.84 ppm). The $^1$H NMR spectrum exhibited signals for four olefinic protons ($\delta$ 7.33, 6.26, 6.13, 5.90) corresponding to two conjugated double bonds, four oxygen-bearing methines ($\delta$ 3.85, 3.69, 3.59, 3.43), five methyls, and other aliphatic protons. The $^{13}$C NMR spectrum indicated the presence of one carbonyl group ($\delta$ 168.7) and one oxygen-bearing quaternary carbon ($\delta$ 103.2, C-11). The COSY spectrum established two partial substructures, which could be connected via a spiral function by analysing HSQC and HMBC correlations (Supplementary Figs. 4–11). Compound **1** was crystallised in methanol solution, and the structure was determined via X-ray crystallography (Supplementary Fig. 12). Therefore, compound **1** was identified as a new *Streptomyces*-derived natural product named pteridic acid H.

Compound **2**, a white solid, is an isomer of **1** deduced by MS with the same molecular formula of $C_{21}H_{34}O_6$. The $^1$H NMR spectrum exhibited signals for four olefinic protons ($\delta$ 7.16, 6.25, 6.07, 5.97) corresponding to two conjugated double bonds, four oxygen-bearing methines ($\delta$ 3.88, 3.66, 3.56, 3.32), five methyls, and other aliphatic protons. The $^{13}$C NMR spectrum indicated the presence of one carbonyl group ($\delta$ 170.2) and one oxygen-bearing quaternary carbon ($\delta$ 103.2, C-11). HSQC and HMBC correlations confirmed a spiroketal skeleton. NOESY spectrum confirmed its relative configurations, where correlations between H-21 and H-13 and H-15, H-7 and H-12a, and H6, H-10, and Me-18 were observed. The key NOESY correlations between H-7 and H-12a revealed a different spiroketal structure than **1** (Supplementary Fig. 11, 13-20). This can be reflected by the relative upfield NMR data for C-12 ($\delta$ 33.8 vs $\delta$ 37.4 in **1**). Compound **2** was identified as pteridic acid F, previously isolated from *Streptomyces pseudoverticillus* YN17707 and a marine-derived *Streptomyces* sp. SCSGAA 0027[25,26].

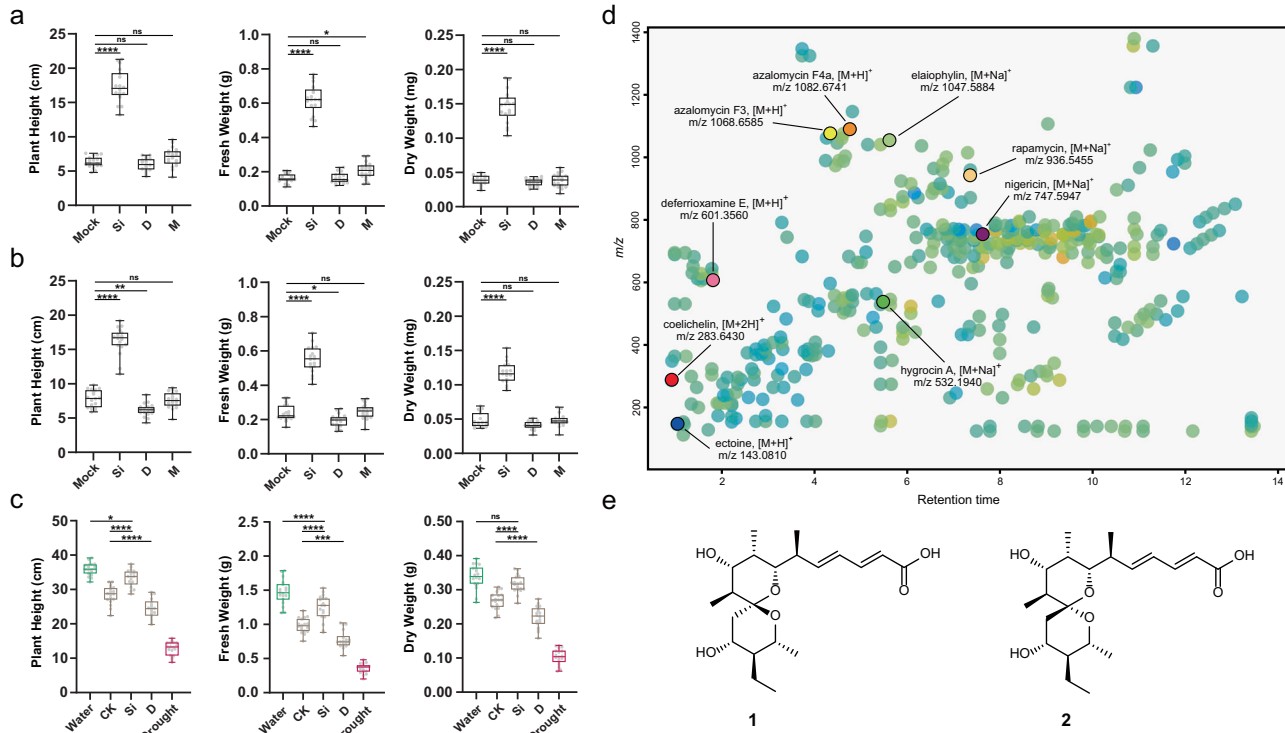

**Fig. 1 | The effects of *S. iranensis* on barley seedlings under abiotic stress and the bioactive components produced by *S. iranensis*. a** The box plots depict the plant height, fresh weight and dry weight of barley seedlings growing under osmotic stress mediated by 20% (w/v) PEG-6000 (mean ± SD, *n* = 18). **b** the box plots depict the plant height, fresh weight and dry weight of barley seedlings growing under salinity stress mediated by 100 mM NaCl (mean ± SD, *n* = 18). In **a** and **b**, statistical significance was assessed by one-way ANOVA with post hoc Dunnett's multiple comparisons test. Asterisks indicate the level of statistical significance: *$p < 0.05$, **$p < 0.01$, ***$p < 0.001$ and ****$p < 0.0001$. Abbreviation: Mock: control, Si: treatment of *S. iranensis* culture broth; D: treatment of *S. iranensis/ΔptaA* culture broth; M: treatment of blank medium (ISP2). **c** The box plots depict the plant height, fresh weight and dry weight of barley seedlings growing under drought stress (mean ± SD, *n* = 18). Different colours of box plots indicate different growing conditions: green, 21 days water; brown, 7 days treatment after 7 days water followed by 7 days drought; red, 7 days water followed by 14 days drought.

Statistical significance was assessed by one-way ANOVA with Tukey test. Asterisks indicate the level of statistical significance: *$p < 0.05$, **$p < 0.01$, ***$p < 0.001$ and ****$p < 0.0001$. Abbreviation: Water: well water for 21 days; CK, 7 days treatment of water after 7 days water + 7 days drought; Si: 7 days treatment of *S. iranensis* culture broth after 7 days water + 7 days drought; D: 7 days treatment of *S. iranensis/ΔptaA* culture broth after 7 days water + 7 days drought; M: treatment of blank medium (ISP2); Drought: 14 days drought after 7 days water. Statistical significance was assessed by one-way ANOVA with Tukey test. **d** The metabolite profile of the native *S. iranensis* growing in liquid ISP2 medium, the known secondary metabolites were identified by HR-LC-MS and highlighted; **e** the bioactive components pteridic acid H (**1**) and pteridic acid F (**2**) isolated from *S. iranensis*. All box plots with centre lines showing the medians, boxes indicating the interquartile range, and whiskers indicating a range of minimum to maximum data beyond the box. Source data are provided as a Source Data file.

## Abiotic stress mitigation of pteridic acids *in planta*

Initially, we tested the effects of different concentrations of pteridic acids H and F on *Arabidopsis* growth in the absence of abiotic stress. A concentration of 0.5 ng ml$^{-1}$ of both pteridic acids H and F was found to significantly promote the growth of *Arabidopsis* seedlings (Supplementary Fig. 21). Under drought stress, pteridic acid H at a concentration of 0.5 ng mL$^{-1}$ increased the root length and fresh weight of *Arabidopsis* seedlings by 54.5% and 89%, respectively, and its activity was significantly better than IAA and ABA at the same molar concentration (Fig. 2a, c). The treatment of pteridic acid F also showed great activity in alleviating drought stress, and the root length and fresh weight were increased by 30.5% and 56.7%, respectively (Fig. 2c). Pteridic acids H and F also showed significant activity in alleviating NaCl-mediated salinity stress (Fig. 2b, d). Compared to the non-treated groups, the treatment of 0.5 ng mL$^{-1}$ pteridic acids H and F increased root length of *Arabidopsis* seedlings by 74.0% and 61.8%, as well as fresh weight by 126.2% and 110.9%, respectively (Fig. 2d).

To get a first understanding how pteridic acids help *Arabidopsis* mitigate salinity stress, we used messenger RNA sequencing (mRNA-seq) to profile the transcripts of *Arabidopsis* seedlings treated with pteridic acid F or H under NaCl-mediated salt stress (Fig. 3a). Compared to the control (CK, treatment with an equal amount of water

under NaCl-mediated salt stress), we observed significant differences in the gene expression patterns upon treatment with pteridic acid (PH or PF) by Pearson correlation analysis (Supplementary Fig. 22). The Differentially Expressed Genes (DEGs) analysis revealed 3575 DEGs (1405 upregulated and 2170 downregulated), and 3727 DEGs (1555 upregulated and 2172 downregulated) in the pteridic acid H treatment versus control (PH vs. CK) and pteridic acid F treatment versus control (PF vs. CK), respectively (Supplementary Data 1). Meanwhile, 1226 upregulated genes and 1860 down-regulated genes were shared between PH and PF treatments. Several abiotic stress-related genes were significantly upregulated after pteridic acids treatments (Fig. 3b), including *SLAC1 HOMOLOGUE 1* (*SLAH1*, AT1G62280)[27], *PIN-FORMED 6* (*PIN6*, AT1G77110)[28], *FERRIC REDUCTION OXIDASE 6* (*FRO6*, AT5G49730)[29] and *TOO MANY MOUTHS* (*TMM*, AT1G80080)[30]. Compared with the control, *ALPHA CARBONIC ANHYDRASE 8* (*ATACA8*, AT5G56330)[31], and *IAA-LEUCINE RESISTANT 2* (*ILR2*, AT3G18485)[32] were uniquely upregulated in the PH treatment samples, while *ATP-BINDING CASSETTE G17* (*ABCG17*, AT3G55100)[33], *DIACYLGLYCEROL KINASE 4* (*DGK4*, AT5G57690)[34] and *MAP KINASE KINASE 7* (*MKK7*, AT1G18350)[35] were upregulated in the PF treatment samples.

Next, we performed the Gene Ontology (GO) enrichment analysis of DEGs focusing primarily on the biological processes

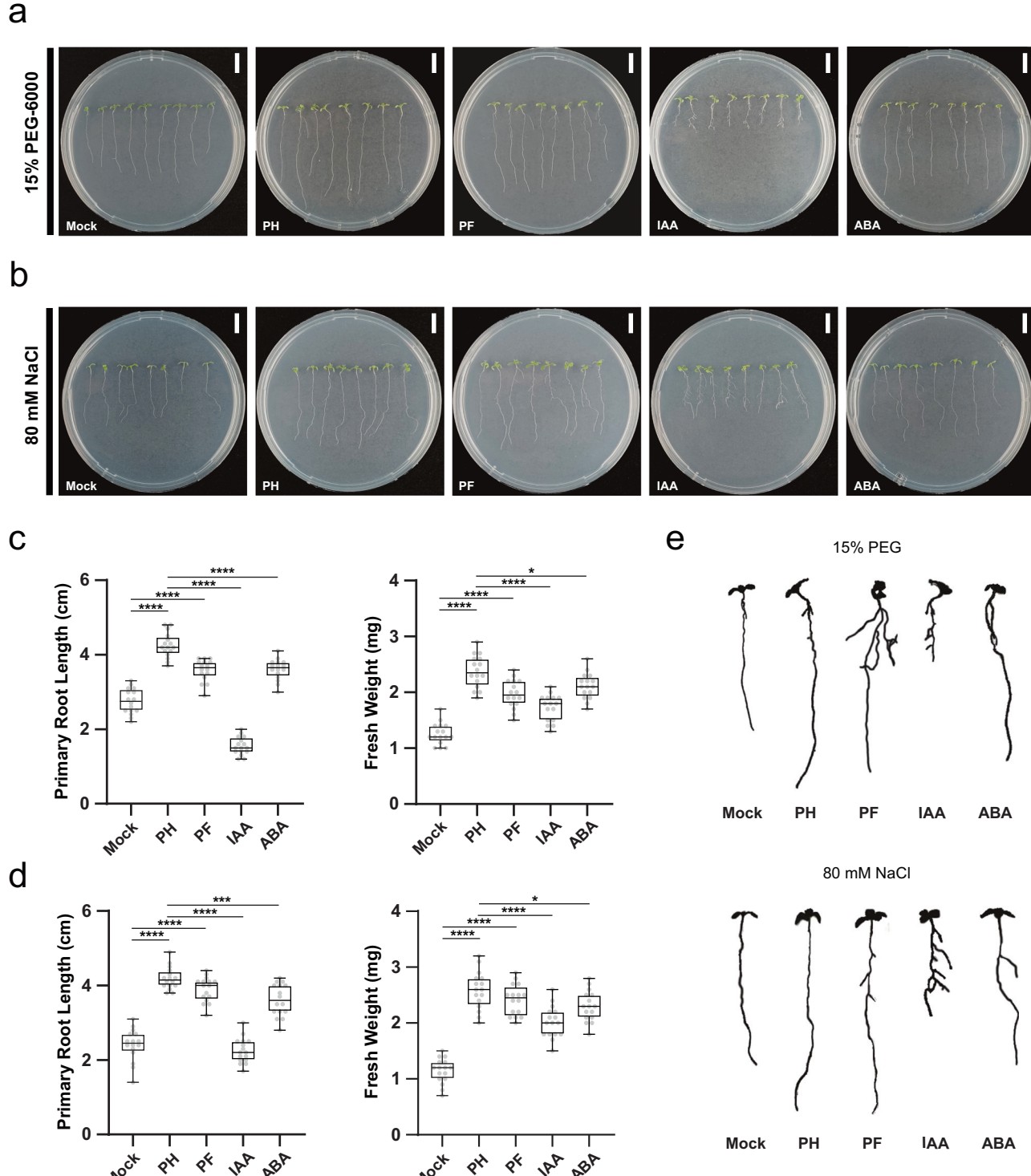

**Fig. 2 | The effect of pteridic acids H and F on *Arabidopsis* seedlings under abiotic stress. a** Phenotype of *Arabidopsis* seedlings growing under drought stress mediated by 15% (w/v) PEG-6000 using other treatments (bars = 1 cm); **b** Phenotype of *Arabidopsis* seedlings growing under salinity stress mediated by 80 mM NaCl using other treatments (bars = 1 cm); **c** The box plots depict the primary root length and fresh weight of *Arabidopsis* seedlings growing on non-stress condition (mean ± SD, *n* = 16); **d** The box plots depict the primary root length and fresh weight of *Arabidopsis* seedlings growing on drought stress condition (mean ± SD, *n* = 16). In **c** and **d**, statistical significance was assessed by one-way ANOVA with Tukey test.

Asterisks indicate the level of statistical significance: *$p < 0.05$, **$p < 0.01$, ***$p < 0.001$ and ****$p < 0.0001$; **e** differences of lateral root growth of *Arabidopsis* seedlings growing in other conditions. Mock: control; PH: treatment of 0.5 ng mL$^{-1}$ pteridic acid H; PF: treatment of 0.5 ng mL$^{-1}$ pteridic acid F; IAA: treatment of 1.3 nM indole-3-acetic acid; ABA: treatment of 1.3 nM abscisic acid. All box plots with centre lines showing the medians, boxes indicating the interquartile range, and whiskers indicating a range of minimum to maximum data beyond the box. Source data are provided as a Source Data file.

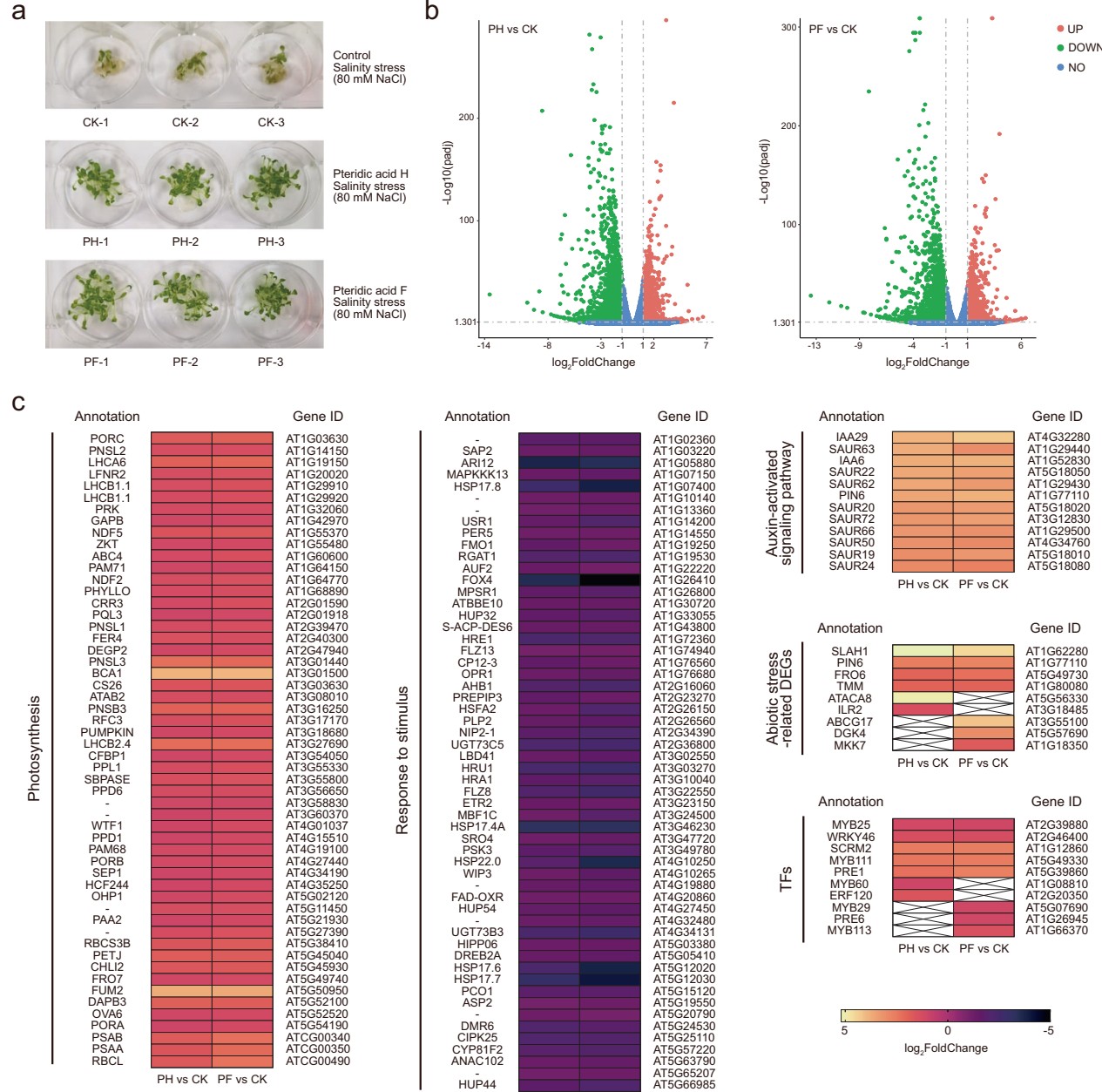

**Fig. 3 | Effect of exogenous pteridic acids on transcription in *Arabidopsis* seedlings under salt stress. a** The phenotype of *Arabidopsis* seedling samples in different treatments under salinity stress. CK: control; PF: pteridic acid F treatment; PH: pteridic acid H treatment. **b** The volcano plots of DEGs identified by mRNA-seq in *Arabidopsis* seedlings treated by pteridic acid H and F under salinity stress. **c** heat-maps of DEGs enriched in photosynthesis, response to stimulus, auxin-activated signalling pathway, abiotic stress defense and TFs.

(Supplementary Data 2). Among all upregulated genes, several highly enriched GO terms such as "Photosynthesis (GO:0015979)", "Plastid organization (GO:0009657)", "Chloroplast organization (GO:0009658)" and "Response to light stimulus (GO:0009416)" suggested pteridic acids could enhance photosynthesis under abiotic stress. For these DEGs with $\log_2$FoldChange $\geq 2$, we observed upregulation of 12 genes belonging to the "auxin-activated signalling pathway (GO:0009734)" (Fig. 3c), suggesting that pteridic acids also trigger auxin-activated signalling transduction. Intriguingly, we observed the downregulations of genes related to "response to stimulus (GO:0050896)", "response to chemical (GO:0042221)" and "response to stress (GO:0006950)", which was speculated to be an interference between different signalling pathway or a negative feedback mechanism of plants to maintain homoeostasis (Fig. 3c)[36]. Moreover,

similar conclusions were reached by the most enriched Kyoto Encyclopedia of Genes and Genomes (KEGG) pathway, pointing to the "Photosynthesis (ath00195)" and "Plant hormone signal transduction (ath04075)" pathways in the *Arabidopsis* seedlings. In addition, the pathway of "Motor proteins (ath04814)", "Ribosome (ath03010)", "Glucosinolate biosynthesis (ath00966)", "Porphyrin metabolism (ath00860)" and "Flavonoid biosynthesis (ath00941)" were also activated, which had been previously reported that these are closely associated with plant abiotic stress resistance[37,38]. On the contrary, pteridic acids downregulated several genes enriched in "Phenylpropanoid biosynthesis (ath00940)", "Cyanoamino acid metabolism (ath00460)", "Glutathione metabolism (ath00480)", "MAPK signalling pathway-plant (ath04016)" and "Starch and sucrose metabolism (ath00500)" pathways.

Transcription Factors (TFs) play an important function in coping with abiotic stress tolerance. Several TFs have been proved to participate in plant salt stress responses, such as AP2/ERF, bZIP and MYB[39–41]. Within these DEGs, we identified various TFs, comprising AP2/ERF (42 unigenes), MYB (35 unigenes), WRKY (27 unigenes), bHLH (31 unigenes) and bZIP families (6 unigenes) (Supplementary Data 3). A handful of plant stress resistance-related TFs were observed, such as *MYB25*, *MYB111*, *WRKY46*, *PRE1* and *SCRM2*[42–46]. TFs associated with resistance to abiotic stress, such as *MYB60* and *ERF120*, exhibited discernible upregulation in the PH treatment samples compared to the control[47,48]. Conversely, *MYB29*, *MYB113* and *PRE6* showed unique upregulation in the PF vs. CK group[49,50].

A previous study suggested that pteridic acids A and B might have a plant growth-promoting effect like IAA and could stimulate the formation of adventitious roots in kidney beans[25]. However, we observed that pteridic acids H and F displayed different IAA-induced phenotypes. Pteridic acids did not exhibit the function to significantly promote lateral root growth of *Arabidopsis* seedlings like IAA (Fig. 2e)[51]. They were also not capable of promoting the formation of adventitious roots in kidney beans, as shown in Supplementary Fig. 23. Except for drought and salinity stress, we also tested the $CuSO_4$-mediated heavy metal stress alleviation activity of pteridic acids on mung beans. The results showed that the 1 ng mL$^{-1}$ pteridic acid H was as effective as ABA in helping mung beans to relieve heavy metal stress (Supplementary Fig. 24). In conclusion, pteridic acids H and F are widely applicable potent plant growth regulators produced by *Streptomyces* to assist plants in coping with different abiotic stress.

## Biosynthesis of pteridic acids

The retro-biosynthesis analysis indicated that pteridic acids could derive from a modular type I polyketide synthase. A putative *pta* BGC was identified in the whole genome sequence of *S. iranensis*, which shows 87% antiSMASH similarity to the BGC of elaiophylin (BGC0000053 in MiBiG database)[52]. The *pta* BGC spans approximately 56 kb and encodes 20 individual biosynthetic genes responsible for the biosynthesis of core polyketide backbones, precursor, glycosylated substituents, transporters and regulators (Supplementary Table 2). The five consecutive Type I polyketide synthase (PKS) encoding genes within the *pta* BGC consist of one loading module and seven extender modules, which are sequentially extended to form a linear polyketide chain by ketosynthase (KS) domain, acyltransferase (AT) domain, acyl carrier protein (ACP), with additional ketoreductase (KR), dehydratase (DH), and enoyl reductase (ER) domains. The substrate specificity predictions for individual AT domains fit well with the structure of pteridic acids (Supplementary Table 3). The last "Asn" residue is absent in the conserved Lys-Ser-Tyr-Asn tetrad of the KR domain in module 3 (PtaB), which is predicted to be inactive (Supplementary Fig. 25). This is consistent with the nonreduced carbonyl group on the α-carbon in module 3. The first DH domain in module 1 is inactive since it does not have a conserved active motif LxxHxxGxxxxP (Supplementary Fig. 26). Following the thioesterase-mediated release of the polyketide chain, the 6,6-spiroketal core structure is likely formed by spontaneous spiroketalisation of the carbonyl group on C11 and the two hydroxyl groups on C17 and C25. Following a loss of $H_2O$, two differentially oriented spirocyclic rings were formed to yield pteridic acids F and H (Fig. 4a). Remarkably, pteridic acid H showed molecular instability under extreme conditions. In the water solution with high temperature (65°C) or acidity (pH = 3), pteridic acid H is transformed into pteridic acid F (Supplementary Fig. 27). A similar spontaneous transformation from (*S*) to (*R*) chirality at the centre of the spiroketal ring was also observed in 6,6-spiroketal avermectin[53].

## CRISPR base editing in *S. iranensis*

To validate the in silico prediction, we utilised the efficient base editing tool CRISPR-cBEST to experimentally confirm the *pta* BGC[54]. As a non-

model *Streptomyces* strain, *S. iranensis* is hard to genetically manipulate through intergeneric conjugation[55]. Therefore, the conjugation process was systematically optimised in this study (Supplementary Fig. 28). The core polyketide synthase *ptaA* was targeted and inactivated by converting a TGG (Trp) codon at position 916 into the stop codon TAA using CRISPR base editing. The editing event was confirmed by PCR amplification and Sanger sequencing of the editing site (Fig. 4c). As expected, the production of both pteridic acids and elaiophylin was abolished in *S. iranensis*/Δ*ptaA* (Fig. 4c). Plant experiments showed that the treatment of *S. iranensis*/Δ*ptaA* fermentation suspension led to the abolishment of the abiotic stress mitigating effects (Fig. 1a–c). To further confirm the *pta* gene cluster, a bacterial artificial chromosome (BAC) library of *S. iranensis* was constructed. BAC-based cross-complementation of *ptaA* in *S. iranensis*/Δ*ptaA* restored the production of pteridic acids and elaiophylin (Supplementary Fig. 29).

Interestingly, based on isotope-labelled precursor feeding and partial cosmid sequencing-based bioinformatics prediction, this BGC has long been inferred to be responsible for the biosynthesis of the antibacterial elaiophylin[56,57]. In 2015, Zhou et al. reported that the thioesterase in the last module catalysed the formation of symmetrical macrodiolide using two units of linear elaiophylin monomeric seco acid (Fig. 4b)[58]. To confirm whether the biosynthesis of pteridic acids is also thioesterase-dependent, site-specific mutations of residues Met-Glu-Asp to Ile-Lys-Asn were introduced into the active sites of the TE domain in vivo (Supplementary Fig. 30)[59]. HR-LC-MS analysis showed that the mutant strain (M2089I + E2090K + D2091N) no longer produced pteridic acids and elaiophylin (Fig. 4d). Hence, we provide additional evidence via in vivo inactivation and site-directed mutagenesis. Co-production of the plant growth-regulating pteridic acids and the antimicrobial elaiophylin through a shared BGC is intriguing and points to possible joint efforts in helping plants cope with both biotic and abiotic stress.

## Geographical distribution of pteridic acid producers

We surveyed available gene cluster family (GCF) data for all bacteria in the BiG-FAM database[60]. We found that the *pta* BGC (GCF_02696) is strictly restricted to the *Streptomyces* genus. In addition, a total of 55 BGCs with high similarity to the *pta* BGC were detected by BiG-SCAPE[61], among a total of 9386 type I polyketides BGCs in 1965 *Streptomyces* from the NCBI assembly database. Through literature supplementation and data dereplication of other reported producers without sequence information, at least 81 *Streptomyces* are known to produce pteridic acids/elaiophylin or have specific *pta* BGC up to date (Supplementary Table 4). Based on the known sampling information, the *pta*-containing *Streptomyces* display a variety of geographic distribution and biological origins (Fig. 5a). We selected two available *Streptomyces* strains (*Streptomyces violaceusniger* Tu 4113 and *Streptomyces rapamycinicus* NRRL 5491) to test the potential plant growth-promoting activity of these potential pteridic acid producers. The HR-LC-MS analysis of both culture broths revealed that they shared similar metabolite profiles, and both produced pteridic acids H and F (Supplementary Fig. 31). Treatment with both culture broths on barley seedlings also exhibited significant plant growth-promoting activities under osmotic, salinity and drought stress (Supplementary Fig. 32). This evidence suggests that this class of *Streptomyces* and its specific secondary metabolite pteridic acids have unique ecological significance involved in plant abiotic stress resistance.

## Phylogeny and evolution of *pta* BGC

To explore the evolutionary clues of pteridic acid producers, 16S rRNA genes were initially used to assess the relatedness of the collected 34 potential producers of pteridic acids with other streptomycetes that do not contain *pta* BGC (Fig. 5b). The results revealed that, except for *Streptomyces albus* DSM 41398 and *Streptomyces* sp. GMR22, and other

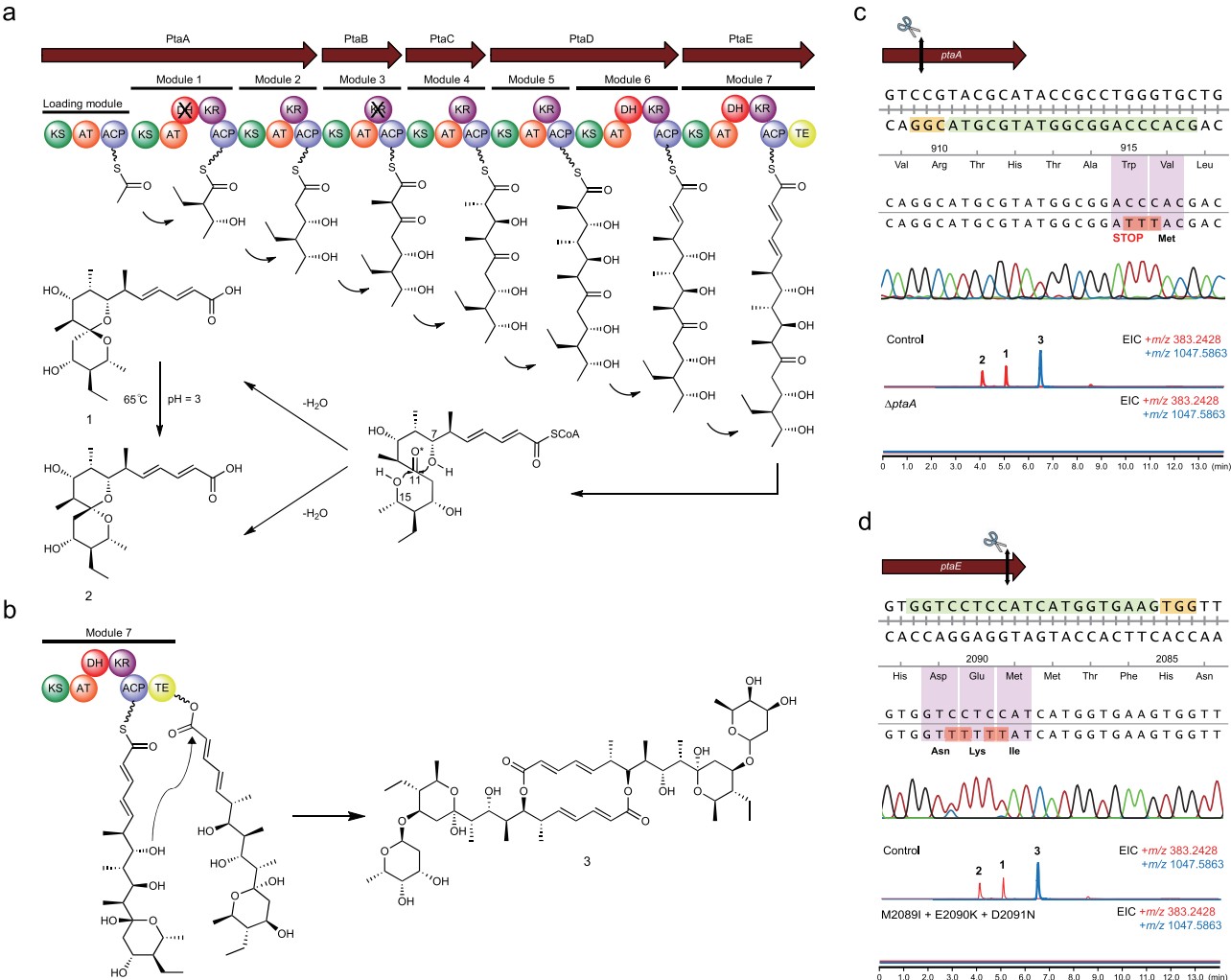

**Fig. 4 | Biosynthesis mechanism of pteridic acids and CRISPR base editing application in *S. iranensis*. a** the proposed biosynthetic pathway of pteridic acids H (**1**) and F (**2**); **b** the proposed biosynthesis mechanism of elaiophylin (**3**), and its macrodiolide formation is catalysed by thioesterase (TE) domain. **c** Sanger sequencing and HR-LC-MS output of CRISPR base editing application of STOP codon introduction targeting the *ptaA* of *S. iranensis*. The 20-nt protospacer sequence is highlighted in light green, whereas the 3-nt PAM sequence is shown in yellow. The codons and corresponding amino acids are indicated, and the black double-headed arrow represents the position of the editing window; Extracted Ion Chromatography (EIC) for **1** and **2** (*m/z* 383.2428 [M + H]⁺) and **3** (*m/z* 1047.5863 [M+Na]⁺) in the wild type *S. iranensis* (Control) and the mutant *S. iranensis*/Δ*ptaA*; **d** Sanger sequencing and HR-LC-MS output of CRISPR base editing application of site-directed mutagenesis targeting the TE domain of *pta* BGC. EIC for **1** and **2** (*m/z* 383.2428 [M + H]⁺) and **3** (*m/z* 1047.5863 [M+Na]⁺) in the wild type *S. iranensis* (Control) and the mutant *S. iranensis*/M2089I + E2090K + D2091N.

*pta*-containing streptomycetes cluster together and are distinct with divergent lineages. To further confirm this hypothesis, two high-resolution *Streptomyces* housekeeping genes, tryptophan synthase subunit beta (*trpB*) and RNA polymerase subunit beta (*rpoB*) were employed to analyse the phylogeny relationship among these strains (Supplementary Fig. 33)[62]. Consequently, only *S. albus* DSM 41398 was classified in a distinct phylogenetic lineage among the *Streptomyces* strains containing *pta* BGC. The strict congruence among the clades of the housekeeping genes indicated dominant vertical transmission and potential horizontal gene transfer of the *pta* BGC in *Streptomyces*.

A total of 15 *pta*-containing streptomycetes with complete genome sequence information were selected to conduct the comparative genomics investigation. The genetic diversity in these streptomycetes was initially revealed using genome sequence similarity analysis. Except for *S. albus* DSM 41398 and *Streptomyces* sp. NA02950, we observed a high degree of similarity in the aligned region, as indicated by both the average nucleotide identity (ANI) and the alignment percentage (AP) among these strains (Supplementary Fig. 34). Genome

synteny analysis revealed that partial genome rearrangements happened among strains even with high sequence similarities (Supplementary Fig. 35). Notably, the *pta* BGC in *S. albus* DSM 41398 (*pta-alb*) is located at the end of the chromosome, a high variable region in *Streptomyces*, suggesting its existence by accepting heterologous biosynthetic gene fragments. The nucleotide sequence alignment of *pta* BGC results showed that *pta-alb* is relatively complete, and the similarity of core genes is proportional to evolutionary relatedness (Fig. 5c). The metabolite profile of *S. albus* DSM 41398 also confirmed the integrity of the *pta-alb* by detecting the production of elaiophylin and pteridic acid H (Supplementary Fig. 36). To further assess the biosynthesis diversity in remaining genetically related strains, we performed the similarity analysis of these BGCs (Supplementary Figs. 37 and 38). The connections between their secondary metabolites, BGCs, revealed that these vertically inherited *Streptomyces* strains also harbour striking similarities. Combining phylogenetic and comparative genomics analysis, we expect that *S. albus* DSM 41398 is evolutionarily the most distinct member from other *pta*-containing streptomycetes and obtained the *pta* BGC via horizontal gene transfer.

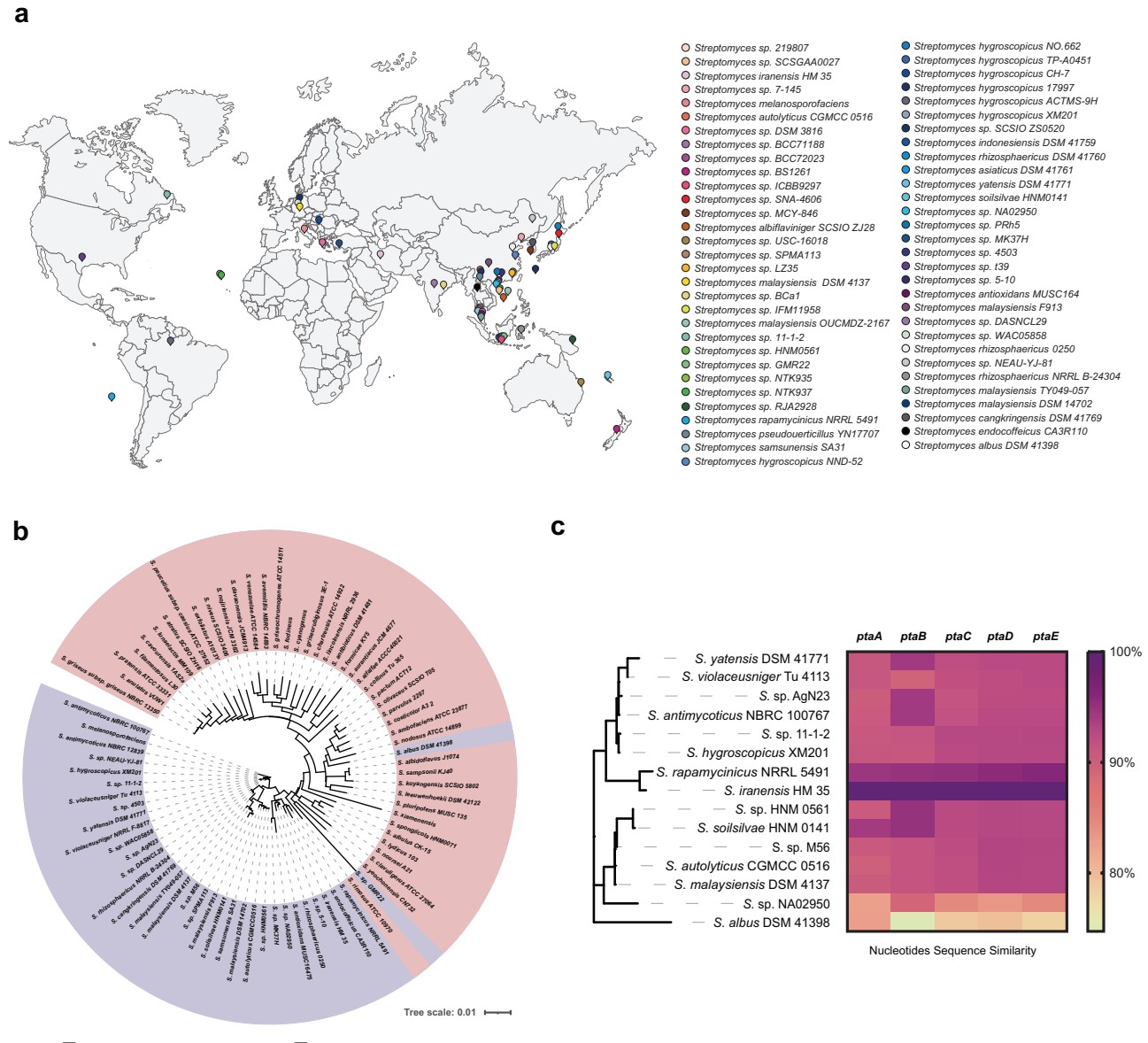

**Fig. 5 | Geographical distribution and phylogenetic analysis of *pta*-containing streptomycetes possessing *pta* BGC. a** A total of 61 streptomycetes were displayed on the map and distinguished by different colours. For detailed strain information, see (Supplementary Table 4). This figure was partly generated using Servier Medical Art (http://smart.servier.com/), licensed under a Creative Common

Attribution 3.0 Generic License; **b** the phylogenetic tree of 16S rRNA nucleotides sequences of *pta*-containing streptomycetes and other streptomycetes; **c** the heatmap depicts similarity differences of core biosynthetic genes of *pta* BGC between *S. iranensis* and other 14 *pta*-containing streptomycetes.

However, most *pta*-containing streptomycetes have vertically inherited *pta* and other BGCs from their ancient ancestors that may be ecologically important and rarely studied.

## Discussion

Drought and salinisation of soil are increasing globally, driving a reduction in crop yields that threatens food security. Plant growth-promoting bacteria is a class of beneficial microorganisms that positively interact with the plant to confer environmental stresses[63]. Although some *Streptomyces* species have been reported to have plant growth-promoting activity, the molecules mediating such positive effects are poorly understood. Deciphering the molecular mechanism is key to understanding the complex plant microbiota interaction. In this study, we present an example of *S. iranensis* secreting a family of secondary metabolites, pteridic acids, to assist plants to cope with abiotic stresses like osmotic, salinity and drought. Pteridic acids H and

F were chemically isolated, structurally characterised and functional validated as plant-beneficial molecules.

Plants respond to harsh environments by changing their physiological processes for better survival[64]. Salt and drought stress signal transduction consists of ionic and osmotic homoeostasis signalling pathways, detoxification (i.e., damage control and repair) response pathways, and pathways for growth regulation[65,66]. Based on mRNA-seq analysis, we identified a total of 3086 DEGs, some of which are associated with diverse abiotic stress defenses in plants. For example, we observed that the upregulation of *SLAH1* is crucial for alleviating the toxicity of salt by root-to-shoot Cl⁻ transport in *Arabidopsis*[27]. *PIN6* was previously identified as the key salt tolerance-related gene in the roots of the mangrove *Avicennia officinalis*[28]. *AtACA8* is a plasma membrane localised Ca²⁺ pump and plays a role in sucrose signalling, ion homeostasis and root development during early seedling germination[31]. *MKK7* positively regulates plant salt tolerance and promotes primary

root growth in *Arabidopsis* seedlings[35]. The differentiation and development of Guard cells (GCs), which are in the epidermis of leaves and stems that regulate stomatal development, are regulated by *TMM*[30]. Under abiotic stress conditions, the upregulation of *ABCG17* expression facilitated the translocation of ABA from the shoot to the root, consequently stimulating lateral root growth[33]. Additionally, our study revealed an enhancement in the lateral root growth of *Arabidopsis* seedlings upon exposure to pteridic acid F, with the expression of the associated *ABCG17* gene observed exclusively in the PF vs. CK group.

Salt stress negatively affects photosynthesis in plants, prompting them to regulate the photosynthetic process, either through intrinsic mechanisms or in response to external stimuli, to enhance their salt tolerance[67]. Based on GO and KEGG pathway enrichment analysis, photosynthesis and its related physiological events of *Arabidopsis* seedlings treated with pteridic acids were immensely upregulated. Interestingly, previous studies also showed that small molecules, such as exogenously applied melatonin, can assist plants in resisting salt stress by improving photosynthesis in different plants[68,69]. Some studies have confirmed that IAA is involved in response to salt stress in plants and a link between IAA signalling and salt stress has been established[70,71]. Herein, we speculated that pteridic acids might also promote plant growth under salt stress via activating auxin signal transduction to promote plant growth under salt stress. B-ARR, as a positive regulator that regulated downstream activity in the cytokinin signalling pathway, was also uniquely upregulated in PF vs. CK group.

Among the various TFs, MYBs participate in various biological processes in plants such as growth, reproduction, secondary metabolism and stress responses[72]. For example, upregulated *MYB25* has been shown to reduce sensitivities toward osmotic and salt stress in *Arabidopsis* and upregulated *MYB111* is a positive regulator of salt stress in *Arabidopsis* by binding directly to the cis-acting element in the promoter region of genes encoding flavonoid synthesis enzymes[42,43]. *MYB60*, which was upregulated in the PH vs. CK group, was previously demonstrated to play a dual role in abiotic stress responses in *Arabidopsis* through its involvement in stomatal regulation and root growth for increased water uptake[47]. *MYB29*, which was uniquely upregulated in the PF vs. CK group, has been demonstrated to be an important factor in promoting *Arabidopsis* lateral root growth under salinity stress[49]. *MYB113* has also been reported to promote anthocyanin biosynthesis in *Arabidopsis* and pear for defense against abiotic and biotic stresses[50]. Previous studies have shown that overexpression of *WRKY46* enhanced root development during salt stress in *Arabidopsis* through modulation of ABA signalling[44]. We found that *WRKY46* was upregulated in pteridic acid-treated samples, which may also serve as a positive regulator in the ABA signalling pathway to confer abiotic stress resistance to plants, although we didn't observe other ABA-related DEGs. The two upregulated bHLH family TFs *PRE1* and *SCRM2* have also been experimentally proved to promote plant growth and resist abiotic stress[45,46].

Horizontal gene transfer is an integral driver of BGC evolution, revealing the independent processes of species phylogeny and BGCs distribution[73]. However, vertical inheritance also influences BGCs evolutionary dynamics, evident from BGCs conservation among closely related strains[74]. We found that the *pta* BGC in *Streptomyces* are widely dispersed geographically and mainly inherited through vertical gene transmission. Some of these strains have also been described to have remarkable biocontrol capabilities. For example, *Streptomyces* sp. AgN23 activates *Arabidopsis* defence responses to fungal pathogen infection by secreting plant elicitors[75], *Streptomyces rhizosphaericus* 0250 and *Streptomyces* sp. 5–10 displayed significant biocontrol potential to fusarium wilt of bitter gourd[76,77]. The family of *pta*-containing *Streptomyces* was also previously described as a specific phylogenetic lineage with the highest BGC abundance and largest genome size across diverse streptomycetes[78]. Although the biosynthetic diversity of these *Streptomyces* strains is likely due to horizontal

transfer events that occurred relatively recently in their evolutionary history instead of genetic diversification through a vertical transfer of BGCs. The multiple Type I PKSs presenting among these strains are highly conserved based on genetic similarity network analysis. There is currently some evidence supporting potential complex cross-BGC regulation in this class of *Streptomyces* strains. Jiang et al. demonstrated that a TetR family transcriptional regulator, *GdmRIII*, controls the biosynthesis of geldanamycin and elaiophylin meanwhile, in *Streptomyces autolyticus* CGMCC 0516[79]. Recently, He et al. found that the rapamycin BGC-situated LAL family regulator RapH co-ordinately regulated the biosynthesis of both rapamycin and elaiophylin in *S. rapamycinicus* NRRL 5491[80]. Although these reports correspond to cross-regulation between evolutionarily conserved BGCs, more details of these communications need to be investigated.

In conclusion, pteridic acids are secondary metabolites produced by streptomycetes enhancing plant resistance to abiotic stress. Transcriptomics profile revealed a higher expression of a diverse set of genes, e.g., in photosynthesis and abiotic stress response genes after pteridic acids treatment. This is a useful illustration of the bacterial metabolite-mediated alteration of plants in response to environmental stress. It will open avenues for utilising *Streptomyces* to rewild plant microbiomes and improve plant abiotic stress resistance to tackle climate change[81].

## Methods

### Strains, plasmids, and cultivation

All strains and plasmids used in this study are listed in (Supplementary Table 5). All *Streptomyces* strains were obtained from the German Collection of Microorganisms and Cell Cultures GmbH (DSMZ, Germany). All *Escherichia coli* strains were grown in liquid/solid LB medium ($5.0\,g\,L^{-1}$ yeast extract, $10.0\,g\,L^{-1}$ peptone, $10.0\,g\,L^{-1}$ NaCl) at 37 °C. All *Streptomyces* strains were grown on SFM medium ($20.0\,g\,L^{-1}$ mannitol, $20.0\,g\,L^{-1}$ soya flour, $20.0\,g\,L^{-1}$ agar), and the SFM medium with the addition of 120 mM calcium chloride solution was used for the step of conjugation at 28 °C. The ISP2 medium ($4.0\,g\,L^{-1}$ yeast extract, $10.0\,g\,L^{-1}$ malt extract, $4.0\,g\,L^{-1}$ dextrose, and 1.0 L distilled water) was used for liquid fermentation of all *Streptomyces* strains used in plant assay and metabolomics analysis. Appropriate antibiotics were supplemented with the following working concentrations: apramycin ($50\,\mu g\,mL^{-1}$), chloramphenicol ($25\,\mu g\,mL^{-1}$), and kanamycin ($50\,\mu g\,mL^{-1}$). All chemicals utilised in this study were from Sigma-Aldrich, USA.

### Metabolomics analyses

High performance liquid chromatography was carried out on the Agilent Infinity 1290 UHPLC system (Agilent Technologies, USA). The 250 × 2.1 mm i.d., 2.7 μm, Poroshell 120 Phenyl Hexyl column (Agilent Technologies, USA) was used for separation. The 2-μL samples were eluted at a flow rate of $0.35\,mL\,min^{-1}$ using a linear gradient from 10% acetonitrile in Milli-Q water buffered with 20 mM formic acid increasing to 100% in 15 min. Each starting condition was held for 3 min before the next run. Mass spectrometry detection was performed on an Agilent 6545 QTOF MS equipped with Agilent Dual Jet Stream electrospray ion source (ESI) with a drying gas temperature of 160 °C, a gas flow of $13\,L\,min^{-1}$, sheath gas temperature of 300 °C, and flow of $16\,L\,min^{-1}$. The capillary voltage was set to 4000 V and the nozzle voltage to 500 V in positive mode. MS spectra were recorded as centroid data at an *m/z* of 100–1700, and auto MS/HRMS fragmentation was performed at three collision energies (10, 20, and 40 eV) on the three most intense precursor peaks per programme. Data were analysed with MassHunter software (Agilent Technologies, USA) and compared with known compounds and crude extract spectral libraries stored in the GNPS platform[45]. The precursor and fragment ion mass tolerance were set as 0.1 Da and 0.02 Da, respectively. In addition, the minpairs cos was set as 0.65, and the minimum matched fragment ions were set as 6.0. The metabolites profile of wild-type *S. iranensis* was visualised by MS-Dial 4.9.2[82].

## Large-scale fermentation and isolation

*S. iranensis* was cultivated in medium 2 (3.0 g L$^{-1}$ CaCl$_2$·2H$_2$O, 1.0 g L$^{-1}$ citric acid/Fe III, 0.2 g L$^{-1}$ MnSO$_4$·H$_2$O, 0.1 g L$^{-1}$ ZnCl$_2$, 0.025 g L$^{-1}$ CuSO$_4$·5H$_2$O, 0.02 g L$^{-1}$ Na$_2$B$_4$O$_7$·10H$_2$O, 0.01 g L$^{-1}$ Na$_2$MoO$_4$·2H$_2$O, and 20.0 g L$^{-1}$ oatmeal in 1.0 L distilled water), at 175 L filling volume in a 300 L fermentation vessel (Sartorius, Germany). The fermentation was carried out for 6 days with aeration of 25-50 L min$^{-1}$, stirring at 200 rpm with a temperature of 28 °C and at a pH range of 5.4-6.4. The fermentation broth was separated, filtered, and loaded onto an Amberchrom CG161Me resin LC column (200 × 20 cm, 6 L). Elution with a linear gradient of H$_2$O-MeOH (from 30% to 100% v/v, flow rate 0.5 L min$^{-1}$, in 58 min) afforded seven fractions (A–G). Fraction G was firstly fractionated by silica gel chromatography with a CH$_2$Cl$_2$/CH$_3$OH gradient to yield 16 fractions, F01-F16. F07 was separated by a Sephadex LH-20 (MeOH) column and twelve sub-fractions F07a-l were obtained. From F07e, **1** (15.0 mg) and **2** (4.0 mg) were obtained by repeated HPLC RP-C$_{18}$ (CH$_3$CN/H$_2$O as gradient).

Pteridic acid H (**1**): white solid; $[\alpha]_D^{20}$ 81 (0.32 mg mL$^{-1}$, CH$_3$OH), $^1$H NMR (800 MHz, MeOD): 7.33 (dd, 15.4 Hz, 11.1 Hz, 1H), 6.26 (dd, 15.2 Hz, 10.8 Hz, 1H), 6.13 (dd, 15.3 Hz, 8.8 Hz, 1H), 5.90 (d, 15.4 Hz, 1H), 3.85 (dd, 10.2 Hz, 2.2 Hz, 1H), 3.69 (overlapping, 1H), 3.59 (m, 1H), 3.43 (m, 1H), 2.48 (m, 1H), 2.29 (dd, 14.9 Hz, 6.1 Hz, 1H), 2.01 (m, 1H), 1.64 (dd, 14.9 Hz, 1.9 Hz, 1H), 1.52 (m, 1H), 1.21 (d, 6.1 Hz, 3H), 1.49 (m), 1.01 (d, 6.8 Hz, 3H), 0.96 (d, 6.8 Hz, 3H), 0.93 (t, 7.3 Hz, 3H), 0.91 (d, 7.0 Hz, 3H); $^{13}$C NMR (200 MHz, MeOD): 168.7, 151.0, 146.7, 129,6, 120.6, 103.2, 75.5, 72.8, 70.3, 70.1, 51.3, 42.2, 40.4, 37.5, 37.4, 24.8, 20.9, 16.0, 12.1, 10.1, 5.0; UV/vis (CH$_3$CN/H$_2$O) $\lambda_{max}$ 262 nm; IR (ATR) $v_{max}$ 2967, 2934, 2879, 1712, 1642, 1600, 1458, 1410, 1383, 1300, 1266, 1223, 1187, 1142, 1109, 1058, 1002, 973 cm$^{-1}$; (+)-HR-ESI-MS (*m/z*) [M + H]$^+$ calcd for C$_{21}$H$_{35}$O$_6$, 383.2428; found, 383.2439. $^1$H NMR and $^{13}$C NMR see Supplementary Tables 6 and 7.

Pteridic acid F (**2**): white solid; $[\alpha]_D^{20}$ −18 (10 mg mL$^{-1}$, CH$_3$OH), $^1$H NMR (800 MHz, MeOD): 5.97 (d, 15.1 Hz, 1H), 7.16 (dd, 15.1 Hz, 10.9 Hz, 1H), 6.25 (dd, 15.1 Hz, 10.9 Hz, 1H), 6.07 (dd, 15.1 Hz, 8.6 Hz, 1H), 3.88 (m, 1H), 3.66 (td, 10.8 Hz, 4.3 Hz, 1H), 3.56 (dd, 11.5 Hz, 4.7 Hz, 1H), 3.32 (m, 1H), 2.49 (m, 1H), 2.19 (dd, 13.1 Hz, 4.3 Hz, 1H), 2.02 (m, 1H), 1.69 (m, 1H), 1.32 (dd, 13.2 Hz, 11.2 Hz, 1H), 1.14 (d, 6.2 Hz, 3H), 1.02 (m, 1H), 1.02 (d, 6.8 Hz, 3H), 0.95 (d, 6.9 Hz, 3H), 0.98 (d, 6.8 Hz, 3H), 1.60 (m, 1H), 1.44 (m, 1H), 0.82 (t, 7.6 Hz, 3H); $^{13}$C NMR (200 MHz, MeOD): 170.2, 148.1, 148.1, 129,6, 122.9, 103.2, 78.0, 74.7, 66.3, 66.3, 52.2, 42.1, 40.5, 38.0, 33.8, 20.4, 15.9, 12.7, 5.3; UV/vis (CH$_3$CN/H$_2$O) $\lambda_{max}$ 264 nm; IR (ATR) $v_{max}$ 2968, 2931, 2877, 1692, 1643, 1618, 1458, 1410, 1380, 1299, 1270, 1188, 1138, 1106, 1059, 1002, 973, 850 cm$^{-1}$; (+)-HR-ESI-MS (*m/z*): [M + H]$^+$ calcd for C$_{21}$H$_{35}$O$_6$, 383.2428; found, 383.2433. $^1$H NMR and $^{13}$C NMR see Supplementary Tables 6 and 7.

## Structure identification

NMR spectra were recorded on an 800 MHz Bruker Avance III spectrometer equipped with a TCI CryoProbe using standard pulse sequences. NMR data were processed using MestReNova 11.0. UHPLC-HRMS was performed on an Agilent Infinity 1290 UHPLC system equipped with a diode array detector. UV-Vis spectra were recorded from 190 to 640 nm. Specific rotations were acquired using Perkin-Elmer 241 polarimeter. IR data were acquired on Bruker Alpha FTIR spectrometer using OPUS version 7.2. TLC analysis was performed on silica gel plates (Sil G/UV$_{254}$, 0.20 mm, Macherey-Nagel). The Biotage Isolera One Flash Chromatography system was used for flash chromatography and performed on silica gel 60 (Merck, 0.04−0.063 mm, 230−400 mesh ASTM). Sephadex LH-20 was from Pharmacia.

## Crystal structure determination

X-ray data collection of **1** was performed on an Agilent Supernova Diffractometer using CuKα radiation. Data were processed and scaled using the CrysAlisPro software (Agilent Technologies, USA). The structure was solved using SHELXS and refined using SHELXL.

Hydrogen atoms were included in ideal positions using riding coordinates. The absolute configuration was determined based on the Flack parameter. Crystal Data for **1**: C$_{21}$H$_{34}$O$_6$, *M* = 382.50, monoclinic, *a* = 8.4619(1) Å, *b* = 15.1661(2) Å, *c* = 8.4994(1) Å, α = 90.00°, β = 107.768(1)°, γ = 90.00°, *V* = 1069.55(2)Å$^3$, *T* = 120.(2) K, space group *P*2$_1$, *Z* = 2, *μ*(Cu Kα) = 0.698 mm$^{-1}$, 17514 reflections collected, 4275 independent reflections ($R_{int}$ = 0.0226, $R_{sigma}$ = 0.0155). The final $R_1$ values were 0.0249 ($I > 2\sigma(I)$). The final $wR_2$ values were 0.0648 ($I > 2\sigma(I)$). The final $R_1$ values were 0.0252 (all data). The final $wR_2$ values were 0.0651 (all data). The goodness of fit on $F^2$ was 1.057. Flack parameter = 0.13(10).

## Genetic manipulation

All primers used were synthesised by IDT (Integrated DNA Technologies, USA) and listed in (Supplementary Table 8). Plasmids and genomic DNA purification, PCR and cloning were conducted according to standard procedures using manufacturer protocols. PCR was performed using OneTaq Quick-Load 2X Master Mix with Standard Buffer (New England Biolabs, USA). DNA assembly was done by using NEBuilder HiFi DNA Assembly Master Mix (New England Biolabs, USA). DNA digestion was performed with FastDigest restriction enzymes (Thermo Fisher Scientific, USA). NucleoSpin Gel and PCR Clean-up Kits (Macherey-Nagel, Germany) were used for DNA clean-up from PCR products and agarose gel extracts. One Shot Mach1 T1 Phage-Resistant Chemically Competent *E. coli* (Thermo Fisher Scientific, USA) was used for cloning. NucleoSpin Plasmid EasyPure Kit (Macherey-Nagel, Germany) was used for plasmid preparation. Sanger sequencing was carried out using a Mix2Seq Kit (Eurofins Scientific, Luxembourg). All DNA manipulation experiments were conducted according to standard procedures using manufacturer protocols.

## Gene inactivation and site-directed mutagenesis

To use the pCRISPR-cBEST for base editing applications, an oligo was designed as Del-ptaA by the online tool CRISPy-web, and the pCRISPR-cBEST plasmid was linearised by *Nco*I. Mixing the linearised pCRISPR-cBEST plasmid and Del-ptaA with the NEBuilder HiFi DNA Assembly Master Mix (New England Biolabs, USA). The linearised pCRISPR-cBEST plasmid was then bridged by Del-ptaA, ending with the desired pCRISPR-cBEST/Δ*ptaA*. Chemically competent *E. coli* were transformed with the recombinant plasmid and confirmed via PCR amplification (programme: 94 °C for 30 s, followed by 30 cycles consisting of 94 °C for 15 s; 54 °C for 15 s; 68 °C for 40 s, and 68 °C for 2 min) and Sanger sequencing. The experimental procedure for site-directed mutagenesis for the TE domain is the same as described above. The *E. coli-Streptomyces* conjugation experiment was conducted according to the modified protocol in this study, and the mutant *Streptomyces* strains were also confirmed by PCR and Sanger sequencing (Eurofins, France).

## Construction of BAC and genetical complementation

The BAC library of *S. iranensis* was constructed using pESAC13-A from Bio S&T (Montreal, Canada). Based on a high-throughput screening method (unpublished), we selected two BACs 1J23 and 6M10 that cross-cover the *ptaA* gene (Supplementary Fig. 28). The selected BAC clones were further confirmed using four sets of primers, including ID-1J23-right-F/R, ID-1J23-left-F/R, ID-6M10-right-F/R, and ID-6M10-left-F/R (Supplementary Table 8). Subsequently, we introduced these two BAC clones 1J23 and 6M10 into *S. iranensis*/Δ*ptaA* (remove the pCRISPR-cBEST/Δ*ptaA* to obtain antibiotics resistance free strain) separately by conjugation. Exconjugants of mutants were further validated by apramycin resistance screening and PCR.

## Enrichment evaluation of *S. iranensis* in rhizosphere soil

The *S. iranensis* (with the apramycin resistance gene) spore suspension was well mixed with fully sterilised soil and was transferred to a 250 mL

flask. The sterilised germinated barley seed was placed in the centre position of soil in flask and grown at 24 ± 2 °C, 8 h dark/16 h light in the growth chamber for 7 days. Samples were collected from soil within 0.1 cm and 3 cm distance from the barley root, with a specification of 0.1 g soil per sample. Then, these samples were transferred to sterilised 1.5 mL Eppendorf tubes and mixed with 500 µL sterilised $H_2O$. 200 µL of each sample was spread evenly over the solid MS medium with the addition of 50 µg mL$^{-1}$ apramycin and grown at 28 °C for 7 days. The number of *Streptomyces* colonies grown on each plate were counted and statistically analysed.

## *Arabidopsis* growth assays
*A. thaliana* ecotype Columbia (Col-0) was used to test the effects of pteridic acids treatment under drought stress mediated by PEG-6000 (Duchefa Biochemie BV) and salinity stress mediated by NaCl (Duchefa Biochemie BV). The modified Murashige & Skoog medium (2.2 g L$^{-1}$ Murashige & Skoog medium including B5 vitamins, 5.0 g L$^{-1}$ of sucrose, 250 mg L$^{-1}$ MES monohydrate, 7.0 g L$^{-1}$ agar, and adjusted pH to 5.7 with KOH) was used in this study. PEG-6000 (15% w/v) was dissolved in water and filtered through 0.2-micron Sartorius Minisart™ Plus Syringe Filters (Fisher Scientific). 50 mL of the filtered solution was overlaid onto the surface of solidified Murashige & Skoog medium. The plates were left for 24 h to diffuse the PEG into the Murashige & Skoog medium. NaCl was added to the medium to a final concentration of 80 mM for the salinity stress alleviation test. The pure compound pteridic acids H and F (0.50 ng mL$^{-1}$), IAA (0.23 ng mL$^{-1}$), and ABA (0.34 ng mL$^{-1}$) were mixed with different media to a final concentration of approximately 1.3 nM and poured into the plates. Seeds were surface sterilised by washing with 70% ethanol for 2 min, then in sterilisation solution (10% bleach) for 1 min by inverting the tubes, and finally washed five times with sterilised water. The seeds were stratified for 2 days at 4 °C in the dark. Sterilised seeds were placed in Petri dishes (approx. 100 seeds per Petri dish) on Murashige & Skoog medium and grown for 3–4 days in the vertical position in a culture chamber at 22 °C under standard long-day conditions (16/8 h light/ dark photoperiod). After three days of growth, seedlings with similar root lengths (7-10 mm) were transferred to square plates containing Murashige & Skoog medium (control) or Murashige & Skoog medium supplemented with 15% (w/v) PEG-6000 or 80 mM NaCl. 16 seedlings were used per replicate for each treatment. The initial position of the plant root tip was marked with a marker. The plants were grown in the vertical position under standard long day conditions (22 °C, 16/8 h light/dark) for 8 days, and then each plate was scanned using Image Scanner. Primary and lateral root lengths and the total plant weight were then scored. The primary and lateral root length measurements were performed by analysing pictures with the Image J software. Fresh weight measurements were estimated using a precision balance. Whole 8-day-old plants grown in the medium were removed using forceps, dried in tissue paper, and then weighed using a precision balance.

## Barley and mung bean growth assays
The pteridic acid streptomycetes producers were tested for their effects on barley cultivars Guld grown in soil. *S. iranensis* HM 35, *S. rapamycinicus* NRRL 5491, and *S. violaceusniger* Tu 4113 were cultivated in ISP2 medium for 7 days at 28 °C. 500 µL culture broth (ca. 3 × 10$^9$ CFU/mL) was added to 100 g sand soil and well mixed. Treated soils were infiltrated with Milli-Q water, 20% PEG-6000 solutions to simulate osmotic stress, and 100 mM NaCl solutions to simulate salinity stress in soil environments. To simulate drought stress during barley growth, the plant was initially watered for 7 days, then subjected to 7 days of drought stress, and finally allowed to recover with various treatments for another 7 days. Barley seeds were rinsed in distilled water and sterilised with 1% sodium hypochlorite for 15 min. They were then washed with distilled water and germinated in distilled water at 24 °C

for 2 days. Barley seeds were planted in each plastic pot (5 cm × 5 cm × 6 cm, six seedlings per pot) supplemented with different treated soil and grown at 24 ± 2 °C, 8 h dark/16 h light in the growth chamber for 7 days. Plant height (cm) was measured as the aerial part of the plant, and the fresh shoot weight (g) and fresh root weight (g) of each seedling were measured separately. Then, the seedlings were dried in the hot-air oven at 70 °C for 6 h to obtain the dry shoot and dry root weights (g). For heavy metal stress experiment, mung beans were pre-germinated and placed on top of the modified Murashige & Skoog medium agar, supplemented with 10 mM $CuSO_4$, with 1.0 ng mL$^{-1}$ pure substances. All mung beans were grown in the dark at 24 ± 2 °C for 4 days.

## Kidney beans growth assays
The seeds of kidney beans (appr. 2 cm in length, from organic farming) were firstly sterilised successively with ethanol (70% v/v) and sodium hypochlorite (5% v/v), each for 2 min and then rinsed with sterile Milli-Q water (three times). The sterilised seeds were cultivated on modified Murashige & Skoog agar plates for 3–4 days. After germination, the seedlings (with 1.5–2-cm-long roots) were soaked in 10 mL aliquots of testing compounds (pteridic acids H and F, 1.0 ng mL$^{-1}$, dissolved in sterile Milli-Q water) in ultra-clear polypropylene containers (ø 34 mm, vol. 20 mL) with polyethylene caps. The control group was treated with 10 mL sterile Milli-Q water. For each treatment, three repetitions (containers) were used, and each repetition included four seedlings. After 24 h, the seeds were transferred into a cut square petri dish, put on the top layer of sandy soil, and then incubated vertically in a growth chamber (24/22 °C, day/night cycle of 16/8 h, 50%, 60%, 70%, 100% of circulated wind velocity for 12 h, 2 h, 2 h, 8 h) for 7 days. Solutions of pteridic acids H and F (2 mL, 1.0 ng mL$^{-1}$ for both) were added separately into corresponding containers, with sterile Milli-Q water as control, and an extra 8 mL of Milli-Q water was added to each petri dish every other day.

## RNA extraction and mRNA-seq
*Arabidopsis* seeds were sterilised, and seven to ten seeds were cultured in each well of a 6-well plate in 2 mL medium containing modified Murashige & Skoog and 30 mM sucrose with 16 h light, at 24 °C in a controlled environment room. After 7 days, seedlings were washed in 20 mL modified Murashige & Skoog liquid medium and then moved to 20 mL of fresh liquid medium containing 80 mM NaCl and cultured in 50 ml E-flasks with 1 ng mL$^{-1}$ of each pteridic acids H or F (or equal amount of water as control). Whole-seedling plant samples were collected after 72 h when significant phenotypic variation occurred between groups. Total RNA was extracted using RNeasy Plant Mini Kit (Qiagen, Germany). mRNA was purified from total RNA using poly-T oligo-attached magnetic beads. After fragmentation, the first strand cDNA was synthesised using random hexamer primers, followed by the second strand cDNA synthesis using dTTP for the non-directional library construction. The reads were generated using an Illumina NovaSeq 6000 (Novogene, UK) with a paired-end 150 bp configuration. DESeq2 was used to estimate DEGs between different treatments with the threshold of FDR-adjusted *p* values ≤ 0.05 and |log2(Fold-Change)|≥1 if there is no additional statement[83]. The online software g:Profiler (http://biit.cs.ut.ee/gprofiler/gost) was used for GO enrichment and clusterProfiler R package was used to test the statistical enrichment of DEGs in KEGG pathways[84,85].

## Bioinformatics analyses
The identification and annotation of all BGCs of *Streptomyces* secondary metabolites were carried out with antiSMASH 6.0[26]. The threshold for similar BGCs was selected as greater than 45% sequence similarity. The BiG-FAM database and BiG-SCAPE software were used to identify the distribution of *pta* gene clusters in different bacteria and generate the pta-gene cluster family similarity

network[60,61]. A cut-off of 0.3 was used as a raw index similarity metric for the BiG-SCAPE analysis. The alignment of 15 *pta*-containing *Streptomyces* genome sequences was performed using the whole-genome alignment plugin of the CLC Genomics Workbench version 22.0.2 (Qiagen). The minimum initial seed length was 15 bp, and the minimum alignment block length was 100 bp. The networks of Big-SCAPE and GNPS analysis were visualised using Cytoscape 3.9. The phylogenetic analysis was conducted by the online multiple sequence alignment tool MAFFT and visualised by iTOL v5[86,87].

### Statistical analysis
Statistical significance was assessed by one-way ANOVA with post hoc Dunnett's multiple comparisons test, one-way ANOVA with Tukey test or t test (see each figure legend). All analyses were performed using GraphPad Prism version 9. $p$ values < 0.05 were considered significant. Asterisks indicate the level of statistical significance: $*p < 0.05$, $**p < 0.01$, $***p < 0.001$, and $****p < 0.0001$. For all relevant figures, source data and exact $p$ values are provided in the Source Data file.

### Reporting summary
Further information on research design is available in the Nature Portfolio Reporting Summary linked to this article.

## Data availability
The gene sequences used in this study were collected by searching National Center for Biotechnology Information (NCBI). The crystal structure data generated in this study have been deposited in the Cambridge Crystallographic Data Centre database under accession code 1984025. The metabolomics data generated in this study have been deposited in MassIVE under accession code MSV000090745 [https://massive.ucsd.edu/ProteoSAFe/dataset.jsp?task=f9aed10a976641d5b8e5e943b67a1953]. The mRNA-seq data were deposited at NCBI Sequence Read Archive (SRA) database with accession code PRJNA1020394. Source data are provided with this paper.

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

## Acknowledgements

We acknowledge Dr. Yaojun Tong for the discussions on CRISPR-BEST. We thank the support from Dr. Charlotte Held Gotfredsen (DTU NMR Centre) and Dr. Aaron John Christian Andersen (DTU Metabolomics Core). We acknowledge Dr. Ting Yang and Dr. Qing Liu for advice in the transcriptomics experiment. We acknowledge financial support from Carlsberg Infrastructure (CF20-0177), Novo Nordisk Foundation Proof of Concept (NNF20OC0062267), DTU Enable Program, InnoExplorer Grant, Innovation Fund Denmark, and Danish National Research Foundation (DNRF137) for support towards the Centre for Microbial Secondary Metabolites (CeMiSt). Z.Y. acknowledges funding from the China Scholarship Council (202004910340). T.W. acknowledges funding from the Novo Nordisk Foundation (NNF20CC0035580). Y.Q. thanks for Novo Nordisk Foundation (NNF22OC0079928).

## Author contributions

Z.Y. and T.W. designed and carried out genetic experiments and bioinformatic analyses. E.S. isolated the metabolites and characterised the structures; L.D. did structure elucidation, compounds crystallisation, and preliminary plant assays; Z.Y., Y.Q., N.C.K. and E.A. performed the plant and mRNA-seq assays; G.P. and M. A. carried out large scale fermentation and downstream processing; P.H. carried out X-ray crystallography and data analysis. All the authors discussed the results and commented on the manuscript.

## Competing interests

A patent related to this research was filed as a PCT application (PCT/EP2022/081635) by Ling Ding on behalf of Technical University of Denmark. All other authors declare no competing interests.
