## [Peer Review File · Nature Communications]

Streptomyces alleviate abiotic stress in plant by producing pteridic acidsREVIEWER COMMENTS

Reviewer #1 (Remarks to the Author):

Gram-positive soil bacteria, *Streptomyces* are of interest as a plant growth-promoting bacteria in agriculture. In this manuscript, Yang et al. identified pteridic acids H and F from a class of *Streptomyces* as a new class of plant growth and stress regulators. The bioassay analysis using a CRISPR-constructed mutant suggested that *S. iranensis* improves plant growth and stress tolerance by secreting pteridic acids.

A major concern of this manuscript is the lack of quantification and statistical analysis of the plant bioassay results (Figures S21, S22, and S23), which are important to justify the authors' conclusions. The authors need to repeat the experiment and quantify the results. The most important claim of this manuscript is that pteridic acids H and F are new potent plant growth regulators not previously reported, which is supported solely by the non-quantified results of superficial bioassay experiments (e.g. Figure S22). The previous study already reported that other isomers, pteridic acids A and B might act as IAA-like plant growth promoters and induce the formation of adventitious roots in kidney beans (See Results and Discussion "Abiotic stress mitigation of pteridic acids in planta"). To obtain solid evidence that pteridic acids F and H exert the bioactivity via a mechanism distinct from that of pteridic acids A and B, the reviewer strongly recommends careful comparison bioassay of pteridic acids A, B, F, and H. Also, more detailed bioactivity analysis is required (e.g. qPCR or RNAseq analysis of hormone or stress responsive genes). It is possible that the effective concentration of each isomer may differ due to differences in affinity for the target protein. Therefore, concentration dependence should be also tested in bioassay experiments.

<Minor comment>

The authors use Dunnett's test for statistical analysis (See Methods). Dunnett's test can be used only when one group is compared with the others (e.g. Mock vs others, see Wikipedia etc). In this respect, the authors use Dunnett's test properly for Figure 1b, 1c, 4b, 4c, and S1b. However, it is not properly used for Figures 2d, 2e, and 2f.

Reviewer #2 (Remarks to the Author):

In this article, Yang et al. investigate the identity and plant effects of a metabolite produced by members of the *Streptomyces*. The authors start with the observation the strain HM 35 benefits plants during osmotic stress treatments, PEG and NaCl administered in both soil and agar plates. I appreciate the diversity of systems used to test for plant benefits but I don't think the authors can call either of these treatments drought (see a recent preprint comparing the transcriptional effects of PEG or salt on drought imposed in a soil-like substrate). A more direct test would be to actually impose drought by withholding water in the soil system used, I'm not sure why the authors would choose to use PEG instead (PEG can also be used as a carbon source for many *Streptomyces*). The authors should also demonstrate whether the benefit of HM 35 (and the others tested), requires colonization and enrichment in plant roots relative to the substrate. Next, the authors use methods, which are not my expertise so I do not have comments in this section, to isolate and identify the metabolites of interest, the polyketides pteridic acid H and F. The authors nicely recapitulate their effects using HM 35 with the individual compounds and provide genetic evidence in HM 35 linking the strain-derived benefits to the *ptaA* gene. The authors then attempt to provide generality to their findings by demonstrating that the *pta* BGC can be found across numerous *Streptomyces* sourced from locations around the world. This is certainly

interesting but I think the authors overstate their results here especially given that no formal analysis was actually performed. Finally, the authors perform a phylogenetic analyses, which tries to build a case that the pta BGC Streptomyces form a monophyletic and closely related clade indicating vertical transmission of this trait. However, I have several problems with this analysis that are described in comments attached to the pdf.

The absence of line numbers and that the manuscript is formatted in double columns made commenting quite irksome. Instead I attach my comments to the pdf itself.

Reviewer #3 (Remarks to the Author):

In this work, Yang and colleagues report on the discovery of a class of molecules (pteridic acids) that alleviate drought and salinity stress in plants. Two molecules, pteridic acid F & H, were isolated from the plant growth promoting bacterium (PGPB) *S. iranensis* and shown to act as plant growth promoters. The main discovery of the paper is the high activity of the pteridic acid F & H in terms of PGP, which were active already at a low concentration of around 1.5 nM. The authors compared the activity of the molecules and showed their improved properties in comparison to e.g. another growth promoter, ABA. These data have been worked out further and are certainly promising. Furthermore, the bioinformatics have been worked out well and it is interesting to see that one BGC produced both pteridic acids and elaiophilin. However, pteridic acids have already been reported as PGP agents, some aspects of the work are rather preliminary and essential controls are missing, in particular the effect of the ptaA mutant on plant growth.

Major comments

1. The pteridic acids are not novel nor is their bioactivity. Pteridic acids A and B have been published (ref 28 in the paper) and shown to act as PGP agents. That means that the novelty of the paper lies primarily in the specific bioactivity of these new compounds. It is therefore important to compare the activity of the pteridic acids F & H to that of A/B. Since the Pteridic acid A/B have been synthesized via total synthesis (Dias & Salles, J. Org. Chem. 2009, 74, 15, 5584–5589) they may just be available.
2. The compounds have PGP activity. However, that does not mean that the PGP activity of *S. iranensis* is entirely due to the production of pteridic acids F & H. After all, the strain appears to be rather gifted in terms of natural products.
3. Following on the previous point, plant experiments have been performed with the wild-type strain of *S. iranensis*. However, the results for the ptaA mutant are missing, why was that strain not included in the soil experiments? That is an essential thing to test. If the authors are correct, then the ptaA mutant should not act as a PGPB. If it does, that would mean that the PGP activity is caused by other natural products produced by the strain.

Other comments

4. I would suggest creating a genetically complemented strain, showing that the introduction of a plasmid expressing ptaA restores production of pteridic acids.
5. The authors speculate on the function of the pteridic acids (page 4, top of column 1) and conclude that the activity reported for A&B differs from that of F&H. Again, comparison of these pteridic acid variants is important. Only limited experiments were done (both by these authors and those of reference 28) and much more information is required to draw any conclusion on the mode of action and the effect on the IAA pathway.

6. Besides pteridic acids, the strain also produces elaiophilin. Elaiophilin looks roughly like a dimer of pteridic acid; see also the proposed biosynthetic pathway in Zhang et al Mar. Drugs 2022, 20(6), 393. Rather than both molecules being produced at the same time, it is at least as likely that the BGC results in either one or the other, depending on perhaps one gene being switched on or off.

Editorial Note: In their review of the first version of this manuscript, reviewer #2 added their comments to the manuscript file. These comments, excluding minor textual revisions, have been copied into this Peer Review File.

Comments in Article File

Comment 1: I'm a bit puzzled as to why HM 35 was selected for study? Past research using this strain doesn't hint at any plant growth promotion. The statement here suggests that HM 35 was part of a large effort to screen many strains? Is this information somewhere in the manuscript?

Comment 2: PEG is usually used with agar plates to create osmotic stress. When using soil, why not impose a real drought and withhold water? This seems more biologically relevant than PEG if one is using a soil based system.

Comment 3: What's the right negative control here? A non-beneficial *Streptomyces*, a commensal bacteria?

Comment 4: Further, did the authors test whether this effect requires colonization?

Comment 5: "To reveal whether it is a universal phenomenon for pteridic acid producers to help plants cope with abiotic stress". This was not performed. The authors simply investigated the presence of this gene across isolates sourced from different locations. Please be more precise about what was actually performed and found.

Comment 6: This isn't really a formal analysis so I don't think the authors can say that the presence of the pta BGC is enriched in coastal habitats. Of the 81 strains listed, many come from non-plant habitats such as marine and termite gut. the pta BGC could have wide ecological relevance entirely unrelated to its activity on plants.

Comment 7: Can the authors be more explicit with exactly how many strains were used for each phylogenetic analysis? If I understand correctly, the first analysis utilizes trpB and rpoB sequence among pta and non-pta producing *Streptomyces* to test for monophyly in pta production. The second analysis uses only pta-producing *Streptomyces* but what is the motivation for this analysis and why are only 15 strains used?

Comment 8: Well if they form a monophyletic clade then why is it so striking that they share similar metabolic repertoires? Plus striking relative to what? There is no statistical test.

Comment 9: perhaps all y axes should be the same scale in d, e, f

Comment 10: I don't think a heatmap is the best visualization tool here. A boxplot would probably work better. Do rows correspond to different replicates? What is the significance test here?

Comment 11: this is not drought, it's osmotic stress

Comment 12: why just 60 strains shown on the map? 62 have location information in Table S4

Comment 13: why only 14 other strains? Also figure c needs much more explanation. At the moment the caption alone does not allow interpretation of what is presented. In fact, the methods section does not explain what is shown in panel c at all either. I don't really have a clue what is being presented here and why it's important.

Comment 14: Shouldn't there also be at least one (preferably many) non-pta producing *Streptomyces* genome in this analysis? At the moment there is no way to tell whether this level of ANI is exclusive among the pta-producers.

Comments in the Supplementary Information file

Comment 1: any quantification in this experiment or the experiments represented in the next 3 supplemental figures?

Response to Reviewers

We would like to thank the reviewers for your constructive comments that allowed us to improve our manuscript. Your time and help is highly appreciated.

We have addressed all comments as outlined below:

Reviewer #1 (Remarks to the Author):

Gram-positive soil bacteria, *Streptomyces* are of interest as a plant growth-promoting bacteria in agriculture. In this manuscript, Yang et al. identified pteridic acids H and F from a class of *Streptomyces* as a new class of plant growth and stress regulators. The bioassay analysis using a CRISPR-constructed mutant suggested that *S. iranensis* improves plant growth and stress tolerance by secreting pteridic acids.

Comment 1: A major concern of this manuscript is the lack of quantification and statistical analysis of the plant bioassay results (Figures S21, S22, and S23), which are important to justify the authors' conclusions. The authors need to repeat the experiment and quantify the results.

Response 1: We have now added the statistical quantification analysis of related figures.

Comment 2: The most important claim of this manuscript is that pteridic acids H and F are new potent plant growth regulators not previously reported, which is supported solely by the non-quantified results of superficial bioassay experiments (e.g. Figure S22). The previous study already reported that other isomers, pteridic acids A and B might act as IAA-like plant growth promoters and induce the formation of adventitious roots in kidney beans (See Results and Discussion “Abiotic stress mitigation of pteridic acids in planta”). To obtain solid evidence that pteridic acids F and H exert bioactivity via a mechanism distinct from that of pteridic acids A and B, the reviewer strongly recommends careful comparison bioassay of pteridic acids A, B, F, and H.

Response 2: We fully agree that it would be great to compare the biological activities of our isolated pteridic acids with reported pteridic acids A-B. In the past years, we have tried the following ways to obtain pteridic acids A and B:

Firstly, we attempted to contact some research groups to obtain pteridic acids A/B. We have contacted Prof. Yasuhiro Igarashi (Pteridic acids A and B, novel plant growth promoters with auxin-like activity from *Streptomyces hygroscopicus* TP-A0451, *J. Antibiot.*, 2002, <https://doi.org/10.7164/antibiotics.55.764>), Prof. Shigefumi Kuwahara (Enantioselective total synthesis of pteridic acid A, *Chem. Commun.*, 2004, <https://doi.org/10.1039/B416309E>), and Prof. Luiz Dias (Total Synthesis of Pteridic Acids A and B, *J. Org. Chem.*, 2009, <https://doi.org/10.1021/jo9010365>). Unfortunately, these research groups have given us feedback that there are no samples left.

Alternatively, based on comparative genomics analysis, we obtained available *pta*-containing *Streptomyces* strains from DSMZ collection (<https://www.dsmz.de/>), including

Streptomyces rapamycinicus NRRL 5491, *Streptomyces violaceusniger* Tu 4113, *Streptomyces melanosporofaciens* DSM 40318, *Streptomyces cangkringensis* DSM 41769, *Streptomyces albus* DSM 41398, and *Streptomyces yatensis* DSM 41771. These strains were fermented in three medium (ISP2, ISP4 and SFM) for 7 days and the culture broths were extracted and analyzed by HR-LC-MS. The results show that no new peak related to pteridic acids A/B was detected. We speculate that the DH domain in module 1 which is responsible for dehydration might be inactive in these strains. Based on above our efforts, it is still challenging to obtain pteridic acids A/B for comparison at present.

Pteridic acids A and B was original reported to act as IAA-like plant growth promoters (<https://doi.org/10.7164/antibiotics.55.764>), However, the experimental data on pteridic acids A/B effect on plant growth are very scarce and limited in that paper. Based on the detailed *Arabidopsis* assays in our manuscript, the effects of pteridic acids F/H and IAA treatment groups on *Arabidopsis* phenotypes were significantly different (Fig. 2a, 2b and 2c).

Comment 3: Also, more detailed bioactivity analysis is required (e.g. qPCR or RNAseq analysis of hormone or stress responsive genes).

Response 3: We now performed a quantitative real-time PCR analysis for detecting the relative expression level of two stress responsive genes: TONOPLAST INTRINSIC PROTEIN 2;3 (TIP2;3) and SALT OVERLY SENSITIVE 1 (SOS1). TIPs represent one family of aquaporins, and they are involved in regulating water flow in response to osmotic challenges like drought or salinity for a plant cell. SOS1 is a plasma membrane Na⁺/H⁺ antiporter, which is also important for plants to alleviate salinity stress. In response to the ionic toxicity triggered by salt stress, plants rely on the Salt Overly Sensitive (SOS) pathway to transport excessive Na⁺ from the cytoplasm to the apoplast, thus ensuring endogenous ionic homeostasis. The overexpression of SOS1 could significantly increase the salt tolerance of *Arabidopsis* plants. Compared to the control, the relative expression level of TIP2;3 was significantly increased following both 1-hour and 24-hour treatments, while the transcript levels of SOS1 was down-regulated at 1-hour while up-regulated at 24-hour (Fig. 1 here, Fig. 2f in revised manuscript). This result indicated that pteridic acids H and F regulated the TIP2;3 and SOS1 to activate stress-resistance responses in plants.

Fig. 1. The relative expression level of TIP2;3 and SOS1 in *Arabidopsis* seedlings under salinity stress were measured after 1 hour and 24 hours of different treatments. Each treatment contains three biological replicates. Statistical significance was assessed by one-way ANOVA with post hoc Dunnett's multiple comparisons test. Asterisks indicate the level of statistical significance: ** $p < 0.01$, **** $p < 0.0001$.

Comment 4: It is possible that the effective concentration of each isomer may differ due to differences in affinity for the target protein. Therefore, concentration dependence should be also tested in bioassay experiments.

Response 4: We initially measured the growth of *Arabidopsis* seedlings with four concentrations of pteridic acid H (0.25 ng/ml, 0.5 ng/ml, 1 ng/ml, 5.0 ng/ml) under no stress conditions (See Fig. 2 here). The concentration of 0.5 ng/ml pteridic acid H and F exhibited a significant growth-promoting effect on *Arabidopsis* seedlings. Therefore, we selected 0.5 ng/ml pteridic acids H/F for further studies. We have now added the above results to lines 140-143 in the revised manuscript and Fig. 21 in the revised Supplementary Materials.

Fig. 2. Primary root length of *Arabidopsis* seedling treated with different concentrations of pteridic acid H and F. Abbreviation: CK, blank control treated by sterile Milli-Q water; PH, treatment with pteridic acid H. Data is (mean \pm SD, n=8). Statistical significance was assessed by one-way ANOVA with post hoc Dunnett's multiple comparisons test. Asterisks indicate the level of statistical significance: * $p < 0.05$, ** $p < 0.01$, *** $p < 0.001$.

<Minor comment>

Comment 5: The authors use Dunnett's test for statistical analysis (See Methods). Dunnett's test can be used only when one group is compared with the others (e.g. Mock vs others, see Wikipedia etc). In this respect, the authors use Dunnett's test properly for Figure 1b, 1c, 4b, 4c, and S1b. However, it is not properly used for Figures 2d, 2e, and 2f.

Response 5: We appreciate the reviewer for pointing out the mistakes in statistical analysis. We have changed to use different test methods for different analyses and describe them in detail in specific figure legends.

Reviewer #2 (Remarks to the Author):

Comment 6: In this article, Yang et al. investigate the identity and plant effects of a metabolite produced by members of the *Streptomyces*. The authors start with the observation the strain HM 35 benefits plants during osmotic stress treatments, PEG and NaCl administered in both

soil and agar plates. I appreciate the diversity of systems used to test for plant benefits but I don't think the authors can call either of these treatments drought (see a recent preprint comparing the transcriptional effects of PEG or salt on drought imposed in a soil-like substrate). A more direct test would be to actually impose drought by withholding water in the soil system used, I'm not sure why the authors would choose to use PEG instead (PEG can also be used as a carbon source for many *Streptomyces*).

Response 6: Polyethylene glycol (PEG) is a synthetic polymer that was used in plant research to simulate drought stress. However, we agree with the reviewer's suggestion that PEG induces osmotic stress rather than drought stress. Therefore, we have now performed the drought experiment in soil-based system. Barley was first watered for 7 days, then applied drought stress for 7 days followed by recovering with different treatments for 7 days. Based on assays of barley growth phenotypes, the treatment of *S. iranensis* culture broth significantly enhanced growth during recovery from drought stress (see Fig. 3a and 3b here). In addition, the treatment of other two phylogenetically related pteridic acids H/F producers *S. rapamycinicus* and *S. violaceusniger* also showed positive effects on plant growth promotion under drought stress in soil-based system (see Fig. 3c here). In addition, we have now modified the description about “PEG-mediated drought stress in soil system” to “PEG-mediated osmotic stress soil system” in the revised manuscript.

Fig. 3. The effect of *S. iranensis* on barley seedlings growth under drought stress. (a) Phenotypes of barley seedlings growth of different groups in soil drought stress. (b) the box plots (middle bar=median, box

limit=upper and lower quartile, extremes=min and max values) depict the plant height, fresh weight and dry weight of barley seedlings growing under different treatment (mean \pm SD, n = 18). Abbreviations: **Water**, treatment of water for 21 days; **CK**, treatment of medium blank as control for 7 days after 7 days water + 7 days drought-stressed; **Si**, treatment of *S. iranensis* for 7 days after 7 days water + 7 days drought-stressed; **D**, treatment of *S. iranensis*/ Δ ptaA for 7 days after 7 days water + 7 days drought-stressed; **Drought**, 21 days drought-stressed; **Sv**, treatment of *S. violaceusniger* for 7 days after 7 days water + 7 days drought-stressed; **Sr**, treatment of *S. rapamycinicus* for 7 days after 7 days water + 7 days drought-stressed. Statistical significance was assessed by one-way ANOVA with post hoc Dunnett's multiple comparisons test. Asterisks indicate the level of statistical significance: * $P < 0.05$, ** $P < 0.01$, *** $P < 0.001$, and **** $P < 0.0001$.

Comment 7: The authors should also demonstrate whether the benefit of HM 35 (and the others tested), requires colonization and enrichment in plant roots relative to the substrate.

Response 7: For colonization analyses, we firstly introduce the non-integrating plasmid pGM1192-RFP (*E. coli-Streptomyces* shuttle vector for mCherry expression) and integrated pSET152-GFP (*E. coli-Streptomyces* shuttle vector for GFP expression, constructed in this study) into wild-type *S. iranensis* and *S. iranensis*/ Δ azaA/ Δ elaA/ Δ nigA (the mutant with core genes responsible for azalomycin, elaiophylin/pteridic acids and nigericin were abolished). The model strain *Streptomyces albus* J1074 was used as a control here. As a result, we get exconjugants of *S. albus* J1074/GFP, *S. iranensis*/GFP, *S. iranensis*/RFP, *S. iranensis*/ Δ azaA/ Δ elaA/ Δ nigA/GFP and *S. iranensis*/ Δ azaA/ Δ elaA/ Δ nigA/RFP. We clearly saw the GFP is well expressed in *S. albus* J1074 (Fig. 4a here). Unfortunately, we could not detect any fluorescent protein expression in the wild-type *S. iranensis* and *S. iranensis*/ Δ azaA/ Δ elaA/ Δ nigA (Fig. 4a here). It prevents us at this stage from looking into the colonization using confocal laser scanning microscopy. We further performed root *Streptomyces* enrichment assay using the spread plate method, which was described at lines 434-442 in the revised paper. Based on the number of colonies grown on the plate, we can preliminarily speculate that *S. iranensis* is not enriched in the roots of plants (Fig. 4b here).

Fig. 4. Colonization and enrichment assays in plant roots. **a**, phenotypes of the *S. albus* J1074/GFP, *S. iranensis*/GFP and *S. iranensis*/ Δ azaA/ Δ elaA/ Δ nigA/GFP under ultraviolet light. GFP is only expressed in *S. albus* J1074. **b**, enrichment evaluation of *S. iranensis* in rhizosphere soil. Statistical significance was assessed by unpaired t test.

Comment 8: Next, the authors use methods, which are not my expertise so I do not have comments in this section, to isolate and identify the metabolites of interest, the polyketides pteridic acid H and F. The authors nicely recapitulate their effects using HM 35 with the individual compounds and provide genetic evidence in HM 35 linking the strain-derived benefits to the *ptaA* gene. The authors then attempt to provide generality to their findings by demonstrating that the *pta* BGC can be found across numerous *Streptomyces* sourced from locations around the world. This is certainly interesting but I think the authors overstate their results here especially given that no formal analysis was actually performed. Finally, the authors perform a phylogenetic analyses, which tries to build a case that the *pta* BGC *Streptomyces* form a monophyletic and closely related clade indicating vertical transmission of this trait. However, I have several problems with this analysis that are described in comments attached to the pdf. The absence of line numbers and that the manuscript is formatted in double columns made commenting quite irksome. Instead I attach my comments to the pdf itself.

Response 8: We feel sorry for the inconvenience of the format brought to the reviewer. In the revised manuscript, we adjust the format to be more suitable for reviewers to read. We are also grateful to the reviewer for providing these comments. We have noticed that the reviewer highlighted specific remarks related to the above comments in the attached PDF. As a result, we will address each comment individually in our subsequent responses.

Comment 9: In the right column of lines 52-60, reviewer has following comments: I'm a bit puzzled as to why HM 35 was selected for study? Past research using this strain doesn't hint at any plant growth promotion. The statement here suggests that HM 35 was part of a large effort to screen many strains? Is this information somewhere in the manuscript?

Response 9: Inspired by the extensive plant-microbiome interaction, we initially selected rhizosphere *Streptomyces* strains like *S. iranensis* (<https://doi.org/10.1099/ijms.0.015339-0>), *S. rapamycinicus*, etc., isolated from special habitats like high salinity and drought, as research objects to explore their potential interactions with plants. Moreover, we have collected a lot of *Streptomyces* from different geographical and environmental sources followed by preliminarily building an in-house high-throughput and automated strain testing platform in our department, which greatly facilitates the discovery of novel plant-beneficial bacteria and secondary metabolites. Since we did not mention the building of high-throughput platforms and strain screening methods in this manuscript, we have now removed the related description of "... was part of a large effort to screen many strains" in the revised manuscript.

Comment 10: In the right column of lines 38-42, reviewer has following comments: Can the authors be more explicit with exactly how many strains were used for each phylogenetic analysis? If I understand correctly, the first analysis utilizes *trpB* and *rpoB* sequence among *pta* and non-*pta* producing *Streptomyces* to test for monophyly in *pta* production. The second analysis uses only *pta*-producing *Streptomyces* but what is the motivation for this analysis and why are only 15 strains used?

Response 10: In the revised manuscript lines 237-238, we clearly stated that we used 16s rRNA, *trpB* and *rpoB* of 34 strains of *pteridic acids Streptomyces* producers to conduct phylogenetic analysis with other non-*pta* producers. We wanted to explore whether the evolution of *pta* gene clusters is convergent with the evolution of *Streptomyces*. In the second analysis, we used 15

strains because only these strains' complete genome sequences were available on NCBI among the above 34 strains of *pteridic acids Streptomyces* producers. In-depth BGCs annotation and analysis of these 15 strains with complete genome sequences aim to find whether these strains also have similarities or other potential relationship in the biosynthesis of secondary metabolites except for *pta* BGC.

Comment 11: In the right column of lines 83-87, reviewer has following comments: PEG is usually used with agar plates to create osmotic stress. When using soil, why not impose a real drought and withhold water? This seems more biologically relevant than PEG if one is using a soil based system.

Response 11: We assume this comment is related to comment 6. As outlined above, we have now performed the barley experiment using soil-based system according to reviewer's suggestion and the details are given in our response 1.

Comment 12: In the left column of lines 149-152, reviewer has following comments: What's the right negative control here? A non-beneficial *Streptomyces*, a commensal bacteria? Further, did the authors test whether this effect requires colonization?

Response 12: The negative control here is to add the same volume of medium blank without *S. iranensis* and its secondary metabolites. We have modified the description in the revised manuscript for clarity. Considering colonization of *Streptomyces*, we tried to introduce fluorescent protein into *S. iranensis* followed by applying confocal laser scanning microscopy analysis. But the non-integrating plasmid pGM1192-RFP and integrated pSET152-GFP were not expressed in our strain, which we had described in detail in our response 7.

Comment 13: In the left column of lines 277-282, reviewer has following comments about "In conclusion, pteridic acids H and F are new, widely applicable potent plant growth regulators produced by *Streptomyces* to assist plants in coping with abiotic stress with an unusual mode of action and have great value for agriculture applications under climate change": what is the unusual mode of action?

Response 13: Pteridic acids A and B was original reported to act as IAA-like plant growth promoters (<https://doi.org/10.7164/antibiotics.55.764>), However, the experimental data on pteridic acids A/B effect on plant growth are very scarce and limited in that paper. Based on the detailed *Arabidopsis* assays in our manuscript, the effects of pteridic acids F/H and IAA treatment groups on *Arabidopsis* phenotypes were significantly different (Fig. 2). Since we have not yet fully completed the study of the mode of action, we have now modified this description and only state that "pteridic acids H and F are widely applicable potent plant growth regulators produced by *Streptomyces* to assist plants in coping with different abiotic stress" at lines 168-170 in the revised manuscript.

Comment 14: In the left column of lines 371-373, reviewer has following comments about "To reveal whether it is a universal phenomenon for pteridic acid producers to help plants cope

with abiotic stress": This was not performed. The authors simply investigated the presence of this gene across isolates sourced from different locations. Please be more precise about what was actually performed and found.

Response 14: We thank the reviewer for raising this point. We realized this description is not suitable here and we have removed it in revised manuscript.

Comment 15: In the left column of lines 389-392, reviewer has following comments about "Many of these strains have been isolated in the coastal high salinity area, and some of these have been described to have remarkable biocontrol capabilities": This isn't really a formal analysis so I don't think the authors can say that the presence of the *pta* BGC is enriched in coastal habitats. Of the 81 strains listed, many come from non-plant habitats such as marine and termite gut. the *pta* BGC could have wide ecological relevance entirely unrelated to its activity on plants.

Response 15: We have modified the description and only state that these *pta*-containing *Streptomyces* strains have a variety of geographical or biological origins.

Comment 16: In the right column of lines 412-414, reviewer has following comments: Well if they form a monophyletic clade then why is it so striking that they share similar metabolic repertoires? Plus striking relative to what? There is no statistical test.

Response 16: Chung and colleagues previously conducted a comprehensive comparative genomics analysis that categorized *S. iranensis* as a distinct *Streptomyces* phylogenetic lineage associated with rugose-ornamented spores (<https://doi.org/10.1128/mSystems.00489-21>). The study revealed that there is a correlation between BGC abundance and the phylogeny of *Streptomyces*, suggesting that the BGC genotypes have been well conserved throughout the evolutionary period to maintain some ecological functions. A monophyletic group F, which includes *S. iranensis*, was found to possess the highest number of BGCs (with an average of 50) and the largest genome (with an average size of 11.5 Mb), as determined through multidimensional analysis, when compared to several other *Streptomyces* species (see Fig. 5 here). Although the biosynthetic diversity of these *Streptomyces* is likely due to horizontal transfer events that occurred relatively recently in their evolutionary history instead of genetic diversification through a vertical transfer of BGCs. The multiple Type I PKSs presenting among these strains are highly conserved based on genetic similarity network analysis. For better explanation, we have now modified the description at lines 305-308 in the revised manuscript.

Fig. 5. *Streptomyces* 16S rRNA phylogeny and BGC abundance. (a) Neighbor-joining tree of 413 *Streptomyces* 16S rRNA sequences with the Jukes-Cantor distance measure. Bootstrap values were calculated from 1,000 bootstrap replications. Different monophyletic clades are highlighted in different colors. (b) BGC abundance analysis of 152 *Streptomyces* genome sequences. BGC abundance stands for the number of BGCs per genome. BGCs in each genome were predicted by antiSMASH with the detection strictness set to “relaxed.” The graph on the left shows the distribution of BGC abundance in each monophyletic clade. The relationship between genome sizes and BGC abundances is shown on the right. (c) 16S rRNA subtree of the monophyletic group F. Only bootstrap values > 50 are shown (<https://doi.org/10.1128/mSystems.00489-21>)

Comment 17: In figure 2d, 2e and 2f, reviewer has following comments: perhaps all y axes should be the same scale in d, e, f.

Response 17: In the revised manuscript, we have now standardized the range of the y-axes for identical experiments and analyses.

Comment 18: In figure 2g, reviewer has following comments: I don't think a heatmap is the best visualization tool here. A boxplot would probably work better. Do rows correspond to different replicates? What is the significance test here?

Response 18: We have now used the histogram with error bars to visualize data in the revised manuscript. Statistical significance was assessed by one-way ANOVA with post hoc Dunnett's multiple comparisons test.

Comment 19: In figure 4, reviewer has following comments: (1) why just 60 strains shown on the map? 62 have location information in Table S4. (2) this is not drought, it's osmotic stress.

Response 19: We thank the reviewer for pointing out this mistake. (1) There are a total of 62 strains were displayed in Fig. 4a, we have now corrected this mistake. (2) We have now

modified the description of “PEG-mediated drought stress in soil system” to “PEG-mediated osmotic stress in soil system” in the revised manuscript.

Comment 20: In figure 5b, reviewer has following comments: Shouldn't there also be at least one (preferably many) non-pta producing *Streptomyces* genome in this analysis? At the moment there is not way to tell whether this level of ANI is exclusive among the pta-producers.

Response 20: In Fig. 5b, we intend to show that the core biosynthetic genes of pteridic acids are highly conserved among different *pta*-containing *Streptomyces* strains. Therefore, we did not include any non-*pta*-containing strains in this analysis.

Comment 21: In figure 5c, reviewer has following comments: why only 14 other strains? Also figure c needs much more explanation. At the moment the caption alone does not allow interpretation of what is presented. In fact, the methods section does not explain what is show in panel c at all either. I don't really have a clue what is being presented here and why it's important.

Response 21: In fig. 5c, we wanted to demonstrate that these *Streptomyces* are not only evolutionarily similar but also have highly similar secondary metabolite biosynthesis capacity. This analysis serves as evidence for the evolutionary vertical transmission of secondary metabolism in this class of *Streptomyces* and potential interactions among different biosynthetic gene clusters. For readers to understand more clearly, we have modified the writing for this part at lines 247-248 in the revised manuscript. These 14 other *pta*-containing *Streptomyces* strains were collected because the complete genome sequences of these strains were available at NCBI (<https://www.ncbi.nlm.nih.gov/genome>). For other *pta*-containing *Streptomyces* strains listed in Table S4, the genome information is incomplete and therefore not suitable for comparative genomics analysis. Considering that this analysis is not the main goal of this study, we have now moved the original Fig. 5c to Supplementary Fig. 36 in the revised Supplementary Materials.

Reviewer #3 (Remarks to the Author):

In this work, Yang and colleagues report on the discovery of a class of molecules (peridic acids) that alleviate drought and salinity stress in plants. Two molecules, pteridic acid F & H, were isolated from the plant growth promoting bacterium (PGPB) *S. iranensis* and shown to act as plant growth promoters. The main discovery of the paper is the high activity of the pteridic acid F & H in terms of PGP, which were active already at a low concentration of around 1.5 nM. The authors compared the activity of the molecules and showed their improved properties in comparison to e.g. another growth promoter, ABA. These data have been worked out further and are certainly promising. Furthermore, the bioinformatics have been worked out well and it is interesting to see that one BGC produced both pteridic acids and elaiophilin. However, pteridic acids have already been reported as PGP agents, some aspects of the work are rather preliminary and essential controls are missing, in particular the effect of the *ptaA* mutant on plant growth.

Major comments

Comment 22: The pteridic acids are not novel nor is their bioactivity. Pteridic acids A and B have been published (ref 28 in the paper) and shown to act as PBP agents. That means that the novelty of the paper lies primarily in the specific bioactivity of these new compounds. It is therefore important to compare the activity of the pteridic acids F&H to that of A/B. Since the Pteridic acid A/B have been synthesized via total synthesis (Dias & Salles, J. Org. Chem. 2009, 74, 15, 5584–5589) they may just be available.

Response 22: We assume this comment is related to comment 2 of reviewer #1. We have tried to get pteridic A and B for comparison, but they are currently unavailable. The details are given in our response 2.

Comment 23: The compounds have PGP activity. However, that does not mean that the PGP activity of *S. iranensis* is entirely due to the production of pteridic acids F & H. After all, the strain appears to be rather gifted in terms of natural products.

Response 23: Yes, *S. iranensis* harbors many secondary metabolite biosynthetic gene clusters and we detected the production of elaiophylin, hygrocinn, azalomycin, rapamycin, *etc.* besides pteridic acids (Fig. 1d). However, most of these known secondary metabolites are irrelevant to plant growth promoting activity. Under bioactivity-guided chemical isolation, we obtained pure pteridic acids H and F from crude extracts of *S. iranensis*. We compared the growth-promoting effects of wild-type *S. iranensis* and *S. iranensis*/ Δ *ptaA* in barley experiment, and the results showed plant growth-promoting activity disappears with *ptaA* inactivation (Fig. 1a, 1b and 1c). We further confirmed that pure pteridic acids H/F can promote growth and assist *Arabidopsis* seedlings to resist abiotic stress (Fig. 2) as well as induce salinity defence-related transcripts (Fig. 1 here). This evidence favorably supports pteridic acids as the bioactive component of growth-promoting effect.

Comment 24: Following on the previous point, plant experiments have been performed with the wild-type strain of *S. iranensis*. However, the results for the *ptaA* mutant are missing, why was that strain not included in the soil experiments? That is an essential thing to test. If the authors are correct, then the *ptaA* mutant should not act as a PGPB. If it does, that would mean that the PGP activity is caused by other natural products produced by the strain.

Response 24: As description of “DSB-free CRISPR base editing in *S. iranensis*” in this manuscript, we constructed the *ptaA* mutant *S. iranensis*/ Δ *ptaA* using CRISPR-cBEST method and follow by the HR-LC-MS analysis which indicated that pteridic acids were no longer produced (Fig. 3c). We also compared the effects of the mutant strain's fermentation broth on barley seedlings under PEG-mediated osmotic stress and NaCl-mediated salinity stress. Experimental results showed that the *S. iranensis*/ Δ *ptaA* could not help barley alleviate abiotic stress (Fig. 1a and 1b, group D: treatment of *S. iranensis*/ Δ *ptaA*) compared to wild-type strain. We further carried out a drought stress in the soil-based system, and we found that *ptaA*-inactivated *S. iranensis* lost its activity of promoting plant growth (new Fig. 1c in updated manuscript).

Other comments

Comment 25: I would suggest creating a genetically complemented strain, showing that the introduction of a plasmid expressing *ptaA* restores production of pteridic acids.

Response 25: Since *ptaA* is coding a multi-domain enzyme complex with a total of 13,608 nt, it is difficult to directly clone the gene by PCR amplification and integrate it into a commonly used complementation expression vector like pGM1190 and pSET152. As an alternative, we construct the genomic Bacterial Artificial Chromosome (BAC) library of *S. iranensis*. Through a high-throughput screening method (unpublished), we selected two plasmids 1J23 and 6M10 that cross-cover the *ptaA* gene (Fig. 4a here). Subsequently, we introduced these two plasmids into *S. iranensis*/ Δ *ptaA* (remove the pCRISPR-cBEST/ Δ *ptaA* to obtain antibiotics resistance free strain) by conjugation. Exconjugants were further validated by apramycin-resistance screening and PCR (Fig. 4b here). HR-LC-MS/MS analysis of the fermentation products of the *S. iranensis*/ Δ *ptaA*/1J23 and *S. iranensis*/ Δ *ptaA*/6M10 showed that the production of pteridic acid H/F and elaiophylin was restored (Fig. 4c here). Based on the above experiments, we can further confirm the function of *pta* gene cluster. We have added the above results to lines 203-206 in the revised manuscript and Supplementary Fig. 28 in the revised Supporting Materials.

Fig. 6. Complementation experiment of *ptaA*-inactivation mutant of *S. iranensis*. (a) location of pESCA13/1J23 and pESCA13/6M10 in genome of *S.iranensis*. (b) apramycin-resistance screening and PCR verification of *S. iranensis*/ Δ *ptaA*/1J23 and *S. iranensis*/ Δ *ptaA*/6M10. (c) Extract Ion Chromatography (EIC)

in positive mode was performed to detect pteridic acid H (1) and pteridic acid F (2) (m/z 383.2428 $[M+H]^+$ $\Delta \pm 5$ ppm) as well as elaiophylin (3) (m/z 1047.5863 $[M+Na]^+$ $\Delta \pm 5$ ppm) in the *S. iranensis*/ Δ *ptaA*/1J23 (trace I and V), *S. iranensis*/ Δ *ptaA*/6M10 (trace II and VI), wild-type *S. iranensis* (trace III and VII), and *S. iranensis*/ Δ *ptaA* (trace IV and VIII).

Comment 26: The authors speculate on the function of the pteridic acids (page 4, top of column 1) and conclude that the activity reported for A&B differs from that of F&H. Again, comparison of these pteridic acid variants is important. Only limited experiments were done (both by these authors and those of reference 28) and much more information is required to draw any conclusion on the mode of action and the effect on the IAA pathway.

Response 26: We have performed quantitative real-time PCR analysis and the details are given in our response 3 of reviewer #2. We measure relative expression level of two plant stress responsive genes: TONOPLAST INTRINSIC PROTEIN 2;3 (TIP2;3) and SALT OVERLY SENSITIVE 1 (SOS1). Compared to the control, the relative expression level of TIP2;3 was significantly increased following both 1-hour and 24-hour treatments, while the transcript levels of SOS1 were down-regulated at 1-hour while up-regulated at 24-hour (Fig. 1 here). This result indicated that pteridic acids H and F regulated the TIP2;3 and SOS1 to activate stress-resistance responses in plants.

Comment 27: Besides pteridic acids, the strain also produces elaiophylin. Elaiophylin looks roughly like a dimer of pteridic acid; see also the proposed biosynthetic pathway in Zhang et al Mar. Drugs 2022, 20(6), 393. Rather than both molecules being produced at the same time, it is at least as likely that the BGC results in either one or the other, depending on perhaps one gene being switched on or off.

Response 27: Most BGCs express only one major group of metabolites. However, it is commonly observed that the same BGC can code for different secondary metabolites. For example, the type I PKS compounds, divergolides A-D produced by *div* BGC (<https://onlinelibrary.wiley.com/doi/10.1002/anie.201006165>); xiamycin and bixiamycins produced by *xia* BGC (<https://onlinelibrary.wiley.com/doi/10.1002/anie.201204087>); thioangucyclines and tetrangomycin produced by *tac* BGC (<https://doi.org/10.1002/anie.202015570>);

Here, bioinformatic analyses suggest that both elaiophylin and pteridic acids share the same PKS backbones (Fig. 4a here). The thioesterase in PtaE was confirmed to catalyze the dimerization and lactonization to form elaiophylin (<https://doi.org/10.1002/anie.201500401>), or a direct hydrolysis released the polyketide chain to form pteridic acids. In our study, we demonstrate that the release of both compounds was released by TE domain in using site-directed mutagenesis (Fig. 3c). In addition, the inactivation and complementation experiments of *ptaA* also proved that the gene cluster produces both elaiophylin and pteridic acids at the same time.

REVIEWER COMMENTS

Reviewer #1 (Remarks to the Author):

The authors added qPCR data for SOS1 and TIP2;3. Why did the authors choose these two genes? There are many other genes that contribute to salt tolerance. The effect of pteridic acids on the gene expression is not remarkable (Figure 2e; significant increase, but only 2- to 2.5-folds). Also, it was reported that SOS1 gene expression is strongly induced by salinity stress itself (Shi et al., 2000 PNAS; Chung et al., 2007 Plant J). Together, the reviewer thinks that the data is not sufficient to conclude that SOS1 and TIP2;3 expression is a main mechanism for pteridic acid-induced salt stress mitigation. In the response to reviewers, the authors mention that the published result about the bioactivity of pteridic acids A and B is scarce. Such claims made for the previous research should be equally applicable to this paper. Why does only two gene expression analysis reveal the action mechanism of pteridic acids H and F? Why don't the authors check auxin- and ABA-responsive gene expression? If the authors do not provide a larger-scale transcriptional analysis, it's better to remove the qPCR data and weak conclusion about the action mechanism of pteridic acids H and F.

Reviewer #3 (Remarks to the Author):

First of all I would like to commend the authors for doing a very thorough job with these revisions. Many of the issues that existed with the previous version have been addressed appropriately. With that, the paper has improved substantially. The major remaining point is the one raised by two referees, namely The comparison between pteridic acids A and B that have been published and shown to act as PBP agents, and compounds F&H. I do sympathise with the authors regarding the (in)availability of the molecules but it remains quite an important issue that remains unresolved. That said, the work is of interest, and many issues have been addressed well.

POINT-BY-POINT RESPONSE TO THE REVIEWERS' COMMENTS

We would like to thank the reviewers for your helpful comments that allowed us to improve our manuscript.

We have addressed all comments as outlined below:

Reviewer #1 (Remarks to the Author):

Q: The authors added qPCR data for SOS1 and TIP2;3. Why did the authors choose these two genes? There are many other genes that contribute to salt tolerance. The effect of pteridic acids on the gene expression is not remarkable (Figure 2e; significant increase, but only 2- to 2.5-folds). Also, it was reported that SOS1 gene expression is strongly induced by salinity stress itself (Shi et al., 2000 PNAS; Chung et al., 2007 Plant J). Together, the reviewer thinks that the data is not sufficient to conclude that SOS1 and TIP2;3 expression is a main mechanism for pteridic acid-induced salt stress mitigation. In the response to reviewers, the authors mention that the published result about the bioactivity of pteridic acids A and B is scarce. Such claims made for the previous research should be equally applicable to this paper. Why does only two gene expression analysis reveal the action mechanism of pteridic acids H and F? Why don't the authors check auxin- and ABA-responsive gene expression? If the authors do not provide a larger-scale transcriptional analysis, it's better to remove the qPCR data and weak conclusion about the action mechanism of pteridic acids H and F.

A: Many thanks for your comments. To get a deeper understanding how pteridic acids help plants mitigate salinity stress, we have carried out transcriptomics analysis of pteridic acids F and H treatment in *Arabidopsis* seedlings under salinity stress. We have replaced the qPCR data with transcriptomics analysis. Please refer to the main text.

Reviewer #3 (Remarks to the Author):

Q: First of all I would like to commend the authors for doing a very thorough job with these revisions. Many of the issues that existed with the previous version have been addressed appropriately. With that, the paper has improved substantially. The major remaining point is the one raised by two referees, namely The comparison between pteridic acids A and B that have been published and shown to act as PBP agents, and compounds F&H. I do sympathise with the authors regarding the (in)availability of the molecules but it remains quite an important issue that remains unresolved. That said, the work is of interest, and many issues have been addressed well.

A: Thanks for your comments! To get a deeper understanding how pteridic acids help plants mitigate salinity stress, we have carried out transcriptomics analysis. Through the data, we observed a similar effect of both pteridic acid H and F in enhancing stress-related gene expression, despite that some gene expression is specific for the individual acid. This may point to the fact that pteridic acid F could induce the lateral root growth of *Arabidopsis*. In the future, it would be interesting to carry out transcriptomics study to compare pteridic acids A-B with pteridic acids F-H.